JCB Journal of Cell Biology

# Ca²⁺ tunneling architecture and function are important for secretion

Raphael J. Courjaret[1,2], Larry E. Wagner II[3], Rahaf R. Ammouri[1], David I. Yule[3], and Khaled Machaca[1,2]

**Ca²⁺ tunneling requires both store-operated Ca²⁺ entry (SOCE) and Ca²⁺ release from the endoplasmic reticulum (ER). Tunneling expands the SOCE microdomain through Ca²⁺ uptake by SERCA into the ER lumen where it diffuses and is released via IP₃ receptors. In this study, using high-resolution imaging, we outline the spatial remodeling of the tunneling machinery (IP₃R1; SERCA; PMCA; and Ano1 as an effector) relative to STIM1 in response to store depletion. We show that these modulators redistribute to distinct subdomains laterally at the plasma membrane (PM) and axially within the cortical ER. To functionally define the role of Ca²⁺ tunneling, we engineered a Ca²⁺ tunneling attenuator (CaTAr) that blocks tunneling without affecting Ca²⁺ release or SOCE. CaTAr inhibits Cl⁻ secretion in sweat gland cells and reduces sweating in vivo in mice, showing that Ca²⁺ tunneling is important physiologically. Collectively our findings argue that Ca²⁺ tunneling is a fundamental Ca²⁺ signaling modality.**

## Introduction

Agonists activate cell surface receptors triggering signaling cascades that allow the cell to respond to environmental cues and coordinate with other cells and tissues to maintain homeostasis. Ca²⁺ signaling is a primary modality downstream of receptors linked to the activation of phospholipase C (PLC), which hydrolyzes the membrane lipid phosphatidylinositol 4,5-bisphosphate (PIP2) producing inositol 1,4,5-trisphosphate (IP₃) and diacylglycerol (DAG) (Berridge, 2016). IP₃ binds to the IP₃ receptor (IP₃R), an ER Ca²⁺ permeable channel, resulting in a transient Ca²⁺ release phase as store Ca²⁺ content is limited. Depletion of endoplasmic reticulum (ER) Ca²⁺ stores activates store-operated Ca²⁺ entry (SOCE), leading to a smaller more sustained Ca²⁺ influx phase (Prakriya and Lewis, 2015). Ca²⁺ release and SOCE are coupled through a third Ca²⁺ signaling modality known as Ca²⁺ tunneling (Fig. 1 A). During tunneling, Ca²⁺ entering the cell through SOCE channels is taken up by the ER Ca²⁺ ATPase (SERCA) into the ER and released through IP₃Rs (Courjaret and Machaca, 2020; Petersen et al., 2017; Taylor and Machaca, 2019). Tunneling spatially expands SOCE signaling as Ca²⁺ diffuses more efficiently within the ER lumen (Allbritton et al., 1992; Choi et al., 2006; Gilabert, 2020; Mogami et al., 1999; Park et al., 2000). Tunneling also modulates the spatial, temporal, and oscillation dynamics of the Ca²⁺ signal (Courjaret et al., 2018; Courjaret and Machaca, 2014, 2020; Petersen et al., 2017; Taylor and Machaca, 2019). Ca²⁺ tunneling was originally described in pancreatic acinar cells (Gerasimenko et al., 2013; Mogami et al., 1997; Petersen et al., 2017) and has more recently been demonstrated in oocytes and HeLa cells, where the tunneled Ca²⁺ signal is mostly cortical and does not reach effectors deep within the cell (Fig. 1 A) (Courjaret et al., 2017, 2018; Courjaret and Machaca, 2014).

SOCE is mediated by two classes of proteins: the STIM resident ER Ca²⁺ sensors (STIM1-2) with lumenal Ca²⁺ binding domains, and the Orai Ca²⁺-selective PM channels (Orai1-3). Store depletion induces a conformational change in STIM1, resulting in its clustering into higher-order oligomers and exposing the SOAR/CAD domain, which binds to and gates Orai1 (Hirve et al., 2018; Park et al., 2009; van Dorp et al., 2021; Yuan et al., 2009). Clustered STIM1 is enriched at ER–PM contact sites (ERPMCS) where it recruits Orai1 by diffusional trapping and gates it open, thus activating SOCE (Hodeify et al., 2015; Hoover and Lewis, 2011; Wu et al., 2014). SOCE is critical for immune cell activation, muscle development, and secretion (Emrich et al., 2022). This is highlighted by the defects observed in patients with mutations in either STIM1 or Orai1—which phenocopy each other—including severe combined immunodeficiency, muscle hypotonia, ectodermal dysplasia, and anhidrosis with dry skin and heat intolerance (Lacruz and Feske, 2015; McCarl et al., 2009).

Given the requirement for a direct physical interaction between STIM1 and Orai1, their colocalization to ERPMCS is essential. ERPMCS are close appositions between the ER and PM

[1]Research Department, Ca²⁺ Signaling Group, Weill Cornell Medicine Qatar, Qatar Foundation, Education City, Qatar; [2]Department of Physiology and Biophysics, Weill Cornell Medicine, New York, NY, USA; [3]Department of Pharmacology and Physiology, University of Rochester Medical Center, Rochester, NY, USA.

Correspondence to Khaled Machaca: khm2002@qatar-med.cornell.edu.

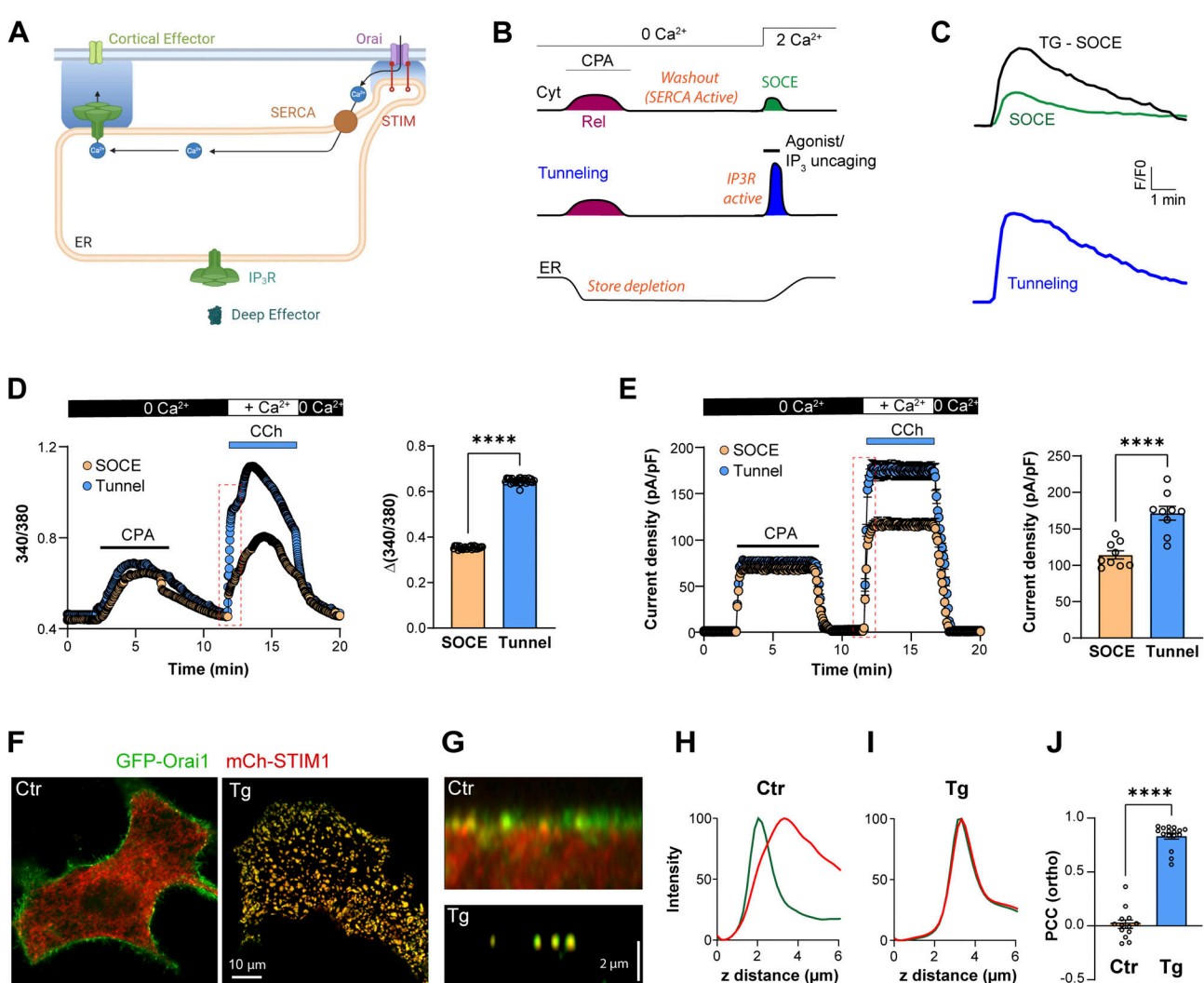

Figure 1. **Ca²⁺ tunneling in primary salivary gland cells. (A)** Cartoon depiction of Ca²⁺ tunneling. STIM1–Orai1 generates localized Ca²⁺ influx within ERPMCS defining the SOCE microdomain. Ca²⁺ taken into the ER by SERCA diffuses through the cortical ER and is released by IP₃R to activate distal cortical targets. Generated using Biorender. **(B)** Comparison of SOCE and Ca²⁺ tunneling protocols. The ER Ca²⁺ stores are depleted by the reversible SERCA inhibitor CPA. This is followed by a washout phase to restore SERCA activity. SOCE develops upon the addition of extracellular Ca²⁺. Concurrent exposure to an agonist to produce IP₃ or uncaging of IP₃ activates Ca²⁺ tunneling. **(C)** Example traces of the cortical Ca²⁺ signals recorded during SOCE, SOCE induced by the irreversible blockade of the SERCA pump by thapsigargin (SOCE-Tg), and during tunneling. Traces are taken from Courjaret et al. (2018). **(D)** Cytosolic Ca²⁺ responses during SOCE and tunneling in isolated salivary gland cells. Tunneling is induced by the addition of CCh (10 μM) simultaneously to Ca²⁺ reperfusion. The bar chart quantifies the peak Ca²⁺ signal of tunneling versus SOCE (n = 26; unpaired t test, P < 0.0001). **(E)** Whole-cell Cl⁻ currents in response to SOCE versus tunneling, showing similar profiles to the Ca²⁺ signals; and bar chart summarizing the data (n = 9; unpaired t test, P < 0.0001). **(F)** Localization at the whole cell level of mCh-STIM1 and GFP-Orai1 in HeLa cells. At rest (Ctr) the confocal plane is in the middle of the cell. After store depletion with thapsigargin (Tg) the image was acquired at the PM optical plane to visualize STIM1-Orai1 co-clusters. **(G)** Orthogonal sections from confocal z-stacks in control conditions and after store depletion. **(H and I)** Relative intensities of STIM1 and Orai1 across z-stack profiles as in G before (Ctr) and after store depletion (Tg). **(J)** Pearson correlation coefficient (PCC) between mCh-STIM and GFP-Orai1 measured in orthogonal slices before (Ctr) and after store depletion (Tg) (n = 13–16; unpaired t test, P < 0.0001).

where the two membranes are <30 nm apart allowing the STIM1 cytoplasmic domain in its extended conformation to span the gap and activate Orai1 (Orci et al., 2009; Shen et al., 2011; Wu et al., 2006). The gap distance between the ER and PM is controlled by different tethering proteins physiologically and can be modulated experimentally using artificial linkers (Fernández-Busnadiego et al., 2015; Giordano et al., 2013; Henry et al., 2022; Várnai et al., 2007). ERPMCS are present at steady state (when Ca²⁺ stores are full) and are stabilized by tethering proteins such as the extended synaptotagmins (E-Syt), TMEM24,

ORPs, and GRAMDs (Chen et al., 2019). Store depletion leads to lateral expansion of ERPMCS (Carreras-Sureda et al., 2023; Henry et al., 2022). Because ERPMCS are the sites of SOCE, their physical dimensions and distribution are critical to understand SOCE signaling and Ca²⁺ tunneling.

The structure of the SOCE microdomain at ERPMCS has been comprehensively discussed by Hogan (Hogan, 2015). The lateral spread of endogenous ERPMCS estimated from electron microscopy studies ranges from 100 to 300 nm and they occupy 1–4% of the total PM area (Hogan, 2015; Orci et al.,

2009; Wu et al., 2006). This limited footprint is coupled to a small number of Orai1 channels within ERPMCS after store depletion, estimated at one to five channels, with a predicted single channel current of ~2 fA and a probability of opening ($P_O$) of ~0.8 (Hogan, 2015; Shen et al., 2021). These estimates argue that SOCE $Ca^{2+}$ signals are spatially confined within ERPMCS to a small percent of the cell cortex. Furthermore, $Ca^{2+}$ diffusion out of the SOCE microdomain at ERPMCS is limited by the high cytosolic $Ca^{2+}$ buffering capacity (Allbritton et al., 1992), as well as $Ca^{2+}$ uptake and extrusion by $Ca^{2+}$ pumps at the PM (PMCA) and ER (SERCA). Measurements of the cortical $Ca^{2+}$ signal due to SOCE alone (when SERCA is active) show a transient low amplitude signal (Fig. 1, B and C, SOCE), which is enhanced in amplitude and duration when SERCA is blocked using thapsigargin (Fig. 1, B and C, TG) (Courjaret et al., 2018). This shows that SERCA-dependent ER $Ca^{2+}$ uptake is important to limit $Ca^{2+}$ diffusion out of the SOCE microdomain. Several studies place SERCA in close proximity to SOCE clusters (Alonso et al., 2012; Basnayake et al., 2021; Courjaret and Machaca, 2014; Jha et al., 2019; Jousset et al., 2007; Manjarrés et al., 2010; Sampieri et al., 2009; Vaca, 2010). PMCA as well has been shown to modulate $Ca^{2+}$ levels in the SOCE microdomain in T cells (Go et al., 2019; Ritchie et al., 2012).

SOCE has been implicated in a broad range of physiological functions (Emrich et al., 2022), which are likely to involve many $Ca^{2+}$-dependent effectors. Some of these effectors localize to the SOCE microdomain and are thus activated directly by $Ca^{2+}$ flowing through SOCE. These include calcineurin, which regulates transcription and immune cell activation through NFAT activation (Kar et al., 2014, 2021), and adenylate cyclase 8, which coordinates crosstalk between SOCE and cAMP signaling (Martin et al., 2009; Willoughby et al., 2012; Zhang et al., 2019). In contrast, other effectors downstream of SOCE, like $Ca^{2+}$-activated channels at the PM, do not localize to the SOCE microdomain and are often excluded from it following store depletion (Courjaret and Machaca, 2014). For such distal effectors, $Ca^{2+}$ tunneling contributes significantly to their activation. These $Ca^{2+}$-activated $Cl^-$ and $K^+$ channels mediate fluid and ion secretion in response to SOCE (Concepcion et al., 2016; Courjaret et al., 2017; Courjaret and Machaca, 2014; Liu et al., 1998). In addition, it is important to note that the $Ca^{2+}$ handling proteins involved in SOCE and tunneling (Orai1, STIM1, PMCA, SERCA, and IP$_3$R) are themselves regulated by $Ca^{2+}$ (transport rate, gating, or conformation).

To better define the physiological role of $Ca^{2+}$ tunneling, we outline its subcellular architecture and test its function *in situ* and *in vivo*. We show that SERCA localizes around the SOCE microdomain at ERPMCS but is excluded from it. IP$_3$R1 localizes more distally from SOCE clusters at an average distance of ~1 μm away. PMCA is diffusely distributed at the PM and overlaps with SOCE clusters. We further generated a novel inhibitor of $Ca^{2+}$ tunneling (CaTAr) by specifically targeting SERCA pumps around the SOCE microdomain. CaTAr blocks tunneling without interfering with either SOCE or IP$_3$-dependent $Ca^{2+}$ release. Blocking tunneling inhibits $Cl^-$ secretion in sweat gland cells and reduces sweating *in vivo* in mice. Collectively, these data show that tunneling is a basic $Ca^{2+}$ signaling modality that is important physiologically for $Cl^-$ and fluid secretion.

## Results

### $Ca^{2+}$ tunneling in primary salivary gland cells

We previously devised a protocol that temporally separates $Ca^{2+}$ tunneling from $Ca^{2+}$ release (Fig. 1 B) (Courjaret et al., 2018; Courjaret and Machaca, 2014, 2020). The $Ca^{2+}$ tunneling signal is significantly larger and more prolonged than the signal due to SOCE alone and is comparable with the $Ca^{2+}$ signal observed following thapsigargin treatment to block ER $Ca^{2+}$ uptake by inhibiting SERCA (Fig. 1 C) (Courjaret et al., 2018). As these experiments were conducted in immortalized cell lines, we wanted to test whether tunneling is operational in a primary cell preparation. We chose acinar cells from the salivary gland as they represent a good model for $Ca^{2+}$ tunneling given their role in saliva secretion and $Ca^{2+}$-dependent $Cl^-$ efflux (Ambudkar, 2018; Concepcion et al., 2016). Agonist-mediated $Ca^{2+}$ signals in salivary acinar cells are largely restricted to the apical membrane where IP$_3$Rs localize within a short distance (50–100 nm) from their target, the $Ca^{2+}$-activated $Cl^-$ channel ANO1 (Sneyd et al., 2021; Takano et al., 2021). The SOCE machinery in contrast localizes predominantly to basolateral membranes (Cheng et al., 2011). Therefore, $Ca^{2+}$ entering through SOCE would have to navigate a considerable distance to activate the $Cl^-$ channels.

Reversible inhibition of SERCA using transient cyclopiazonic acid (CPA) application increases cytosolic $Ca^{2+}$, activates the $Cl^-$ currents, and effectively depletes ER $Ca^{2+}$ as validated by the lack of $Ca^{2+}$ release in response to carbachol (CCh) (Fig. S1 A). To assess the contribution of tunneling in salivary acinar cells, we measured $Ca^{2+}$ signals and $Cl^-$ currents in response to SOCE alone versus tunneling. In both cases, stores were depleted using transient CPA exposure, followed by a washout period to relieve SERCA inhibition (Fig. 1 B). For SOCE alone, we perfused with a $Ca^{2+}$-containing solution, whereas to induce tunneling, we perfused with $Ca^{2+}$ and CCh to activate both SOCE and IP$_3$Rs. The tunneling protocol produced larger $Ca^{2+}$ signals than SOCE alone (Fig. 1 D), which were associated with greater $Ca^{2+}$-activated $Cl^-$ currents (Fig. 1 E). Analyzing the kinetics of the initial rising phase of both the $Ca^{2+}$ (Fig. S1 B) and $Cl^-$ signals (Fig. S1 C) reveals that both develop significantly faster during the tunneling protocol compared with SOCE (7.4-fold for $Ca^{2+}$ and 1.7-fold for $Cl^-$). This shows that tunneling is faster and more efficient in activating ANO1 channels in salivary acinar cells as compared with SOCE alone.

### Architecture of the tunneling machinery

#### STIM1 and Orai1

To better define tunneling mechanistically, we were interested in mapping the architecture of the tunneling machinery. We used STIM1–Orai1 clusters as our reference as they have been studied extensively. STIM1 and Orai1 are diffuse throughout the ER and PM, respectively, at rest (Fig. 1, F and G, Ctr). Store depletion leads to their co-clustering at ERPMCS, which appears as defined puncta in Airyscan images both in the x/y (lateral) (Fig. 1 F) and z (axial) dimensions (Fig. 1 G). Before store depletion, the localization of STIM1 to the ER and Orai1 to the PM are apparent in orthogonal sections (Fig. 1 G) and z profiles (Fig. 1 H). Store depletion leads to the colocalization of STIM1

and Orai1 to ERPMCS, which is the same z plane in Airyscan images (Fig. 1, G and I). In the super-resolution Airyscan mode, the x/y resolution is estimated to be 120 nm whereas the z resolution is poorer at 350 nm (Wu and Hammer, 2021). So fluorescent objects in the 350 nm Airyscan axial plane cannot be resolved. Therefore, for these experiments, we used the peaks from the x/y (Fig. 1 H) and z (Fig. 1 I) profiles to localize the different tunneling effectors relative to each other, primarily using STIM1 as the reference for the SOCE microdomain at ERPMCS.

We calculated the Pearson colocalization coefficient (PCC) between STIM1 and Orai1 in the orthogonal plane. STIM1 and Orai1 are separate under resting conditions (PCC 0.016 ± 0.04) and significantly colocalize following store depletion (PCC 0.83 ± 0.03) (Fig. 1 J). At the cluster level, line scans along the membrane plane through the clusters, and in the axial dimension indicate that the position of STIM1 and Orai1 at our imaging resolution is indistinguishable in x/y and z dimensions (Fig. S1, D–F).

### STIM1 and SERCA

We next localized the ER Ca²⁺ pump SERCA2b relative to SOCE clusters by immunostaining for endogenous STIM1 and SERCA2b. Whenever possible, based on the availability of validated antibodies, we localized endogenous proteins to avoid any artifacts due to overexpression or tagging. At rest, both STIM1 and SERCA are diffusely distributed within the ER (Fig. 2 A, Ctr) and colocalize as indicated in the high magnification image (Fig. 2 B) and line scan through the ER cisterna (Fig. 2 C). Interestingly, store depletion separates SERCA from STIM1 in the x/y axis; as observed at high magnification of a STIM1 cluster surrounded by more diffusely distributed SERCA (Fig. 2 D), and through a line scan profile (Fig. 2 E). Quantifying the peak-to-peak distance between STIM1 and SERCA in the lateral dimension confirms that SERCA, in contrast to Orai1, does not localize to the STIM1 cluster (Fig. 2 F). The PCC between endogenous STIM1 and SERCA2b at the PM focal plane shows a significant reduction in their colocalization following store depletion (Fig. 2 G). A similar exclusion of endogenous SERCA2b was observed from expressed mCh-STIM1 (Fig. S2, A and B).

To assess the distribution of SERCA in the axial dimension, we quantified intensity profiles through the STIM1 cluster (inside) or in its proximity (outside) (Fig. 2 H). A virtual line scan through the STIM1 cluster shows that SERCA is excluded from the cluster and peaks deeper within the ER (Fig. 2 H, Inside). Line scan just outside the STIM1 cluster shows that SERCA localizes axially with the remaining lower intensity unclustered STIM1 deeper within the ER (Fig. 2 H, Outside). These line scans were obtained from the same orthogonal z-slice and normalized to the maximal intensity for STIM1 and SERCA within the slice. Comparative quantification of the axial distance between STIM1 and SERCA shows that in the axis of the STIM1 cluster (in cluster), SERCA localizes deeper than STIM1 as compared to the colocalization of STIM1 and Orai1 in the z-axis (Fig. 2 I). Outside the STIM1 cluster, the distance from the STIM1 peak to SERCA was similar to the STIM1–Orai1 distance, indicating that SERCA localizes in close proximity to the PM but is excluded from SOCE clusters (Fig. 2 I). 3D reconstruction of

STIM1 and SERCA distribution within the cortical ER supports a spatial organization where the STIM1 cluster is surrounded by SERCA, without SERCA localizing to the cluster (Fig. 2 J). Finally, we visualized STIM1 clusters and SERCA at the edge of the cell (in the x/y plane to improve spatial resolution), which also displays a clear separation between STIM1 clusters and SERCA (Fig. S2 C). Collectively, these findings argue that following store depletion, SERCA is not present within STIM1 clusters, but rather localizes cortically in their vicinity, both laterally and axially (Fig. S2 D). Such SERCA distribution is ideally suited to support tunneling as it would not interfere with Ca²⁺ transients within the SOCE microdomain (ERPMCS gap), but would transfer Ca²⁺ leaking out of the microdomain into the ER to fuel tunneling.

### STIM1 and IP₃R

Thillaiappan et al. showed that a population of cortical immobile IP₃Rs ("licensed") initiate Ca²⁺ release signals and that they associate with the actin-binding protein KRAP (Thillaiappan et al., 2017, 2021). This localization, as we have previously argued, seamlessly supports Ca²⁺ tunneling (Taylor and Machaca, 2019). To assess the spatial relationship between STIM1 and IP₃R1, we localized endogenous IP₃R1 and KRAP relative to STIM1 clusters. At rest, STIM1 and IP₃R1 are diffuse through the ER although licensed IP₃R1s are visible as cortical clusters that colocalize with KRAP and appear to align along actin tracks (Fig. S3, A and B). Store depletion leads to STIM1 localization to ERPMCS without a sizable change in IP₃R1 distribution, resulting in significant separation between STIM1 clusters and licensed IP₃R1s (Fig. 2 K). The licensed IP₃R1 clusters colocalize perfectly with KRAP (Fig. 2 K). PCC analyses confirm the colocalization of IP₃R1 and KRAP and the separation of STIM1 clusters from licensed IP₃R1s or KRAP (Fig. 2 L). TIRF imaging replicates the distribution of STIM1, KRAP, and IP₃R1 relative to each other (Fig. S4, A and B), supporting their cortical localization. Similar results were obtained by analyzing the colocalization of IP₃R1–GFP with KRAP or STIM1–Ch (Fig. S4, C–I). IP₃R1 did not colocalize with the STIM1 clusters laterally, although in the z-axis licensed IP₃R1s were in the same focal plane as STIM1, while "mobile" IP₃Rs localized deeper (Fig. S4, K and L). Of note, the expressed GFP-tagged IP₃R1 localized closer to STIM1 clusters (Fig. S4) compared with endogenous IP₃R1 (Fig. 2 K), arguing that either the GFP tag and/or overexpression interfere somehow with IP₃R1 localization.

As IP₃Rs are the site of Ca²⁺ release during Ca²⁺ tunneling, the distance between IP₃R1 and STIM1 after store depletion provides a ruler for the spatial extent of tunneling. We thus quantified the distance between STIM1 clusters and licensed IP₃R1s (employing both IP₃R1 or KRAP) after store depletion using near neighbor distance (NND) analyses. This shows that the majority of IP₃R1s are within 0.2–1.8 µm away from SOCE clusters (Fig. 2 M), with an average distance of 0.98 ± 0.02 µm (Fig. 2 N). The closest IP₃R clusters were within 200 nm from the SOCE microdomain and the farthest were 2.6 µm or more away (Fig. 2 M). These results argue that tunneling extends the SOCE microdomain laterally by up to 10-folds from ~100 to 300 nm estimated SOCE microdomain size (ERPMCS) to a Ca²⁺ signal through IP₃Rs up to 2.6

Figure 2. **Localization of SERCA2b and IP₃R1. (A)** Immunostaining of SERCA2b and STIM1 at the whole cell level before (Control) and after store depletion with Thapsigragin. **(B and D)** High magnification views of the spatial organization of STIM1 and SERCA2b before (Ctr) and after store depletion (Tg) in the lateral dimension (x/y). Scale bar 500 nm. **(C and E)** Intensity profiles were obtained from virtual line scans in the x/y plane across ER tubules (C) and a STIM1

cluster (E). **(F)** Distance between the peak signals of mCh-STIM1 (center of the cluster) and the nearest SERCA2b maximum after store depletion as in panel D detected by immunofluorescence. The distance to the peak of GFP-Orai1 was used as a reference ($n$ = 11–16; unpaired $t$ test, P < 0.0001). **(G)** Pearson's correlation coefficient (PCC) obtained before (Ctr) and after store depletion (Tg) between endogenous SERCA2b and endogenous STIM1. Colocalization at rest was measured in the middle of the cell and at the PM plane after store depletion ($n$ = 9–11; unpaired $t$ test, P = 0.0011). **(H)** Orthogonal section across the PM after store depletion illustrating the relative position and intensity of the STIM1 cluster (red) and SERCA2b (green). The arrows indicate the positions of the line analysis inside the cluster (I) and outside (O). The STIM1 and SERCA2b normalized intensities were measured over the z-axis after store depletion from inside and outside a STIM1 cluster. Scale bar 500 nm. **(I)** Quantification of the peak-to-peak distance in the z-axis between mCh-STIM1, Orai-GFP, and SERCA2b, inside and outside the STIM1 clusters ($n$ = 13–16; one-way ANOVA, P < 0.0001). **(J)** 3D reconstruction of the relative localization of SERCA2b around the STIM1 cluster. **(K)** Top: Immunofluorescence for endogenous STIM1 (red) and IP$_3$R1 (green) after store depletion with Thapsigargin. A virtual line scan (arrow) across the high-magnification image shows the separation between the SOCE cluster and "licensed" IP$_3$R1s. Bottom: Immunofluorescence for endogenous IP$_3$R1s and KRAP at the PM focal plane. The intensity profile measured along the white arrow in the high magnification image indicates the high degree of colocalization of the "licensed" IP$_3$R1 and KRAP. **(L)** PCC summaries between STIM1, IP$_3$R1, and KRAP ($n$ = 8–14; one way ANOVA, P < 0.0001). **(M)** Histogram of the relative frequency of the nearest neighbor distance (NND) between STIM1 clusters and IP$_3$R1 or KRAP (Outliers removed using the ROUT method and a Q value of 10%). **(N)** Violin plot comparing the distribution of NNDs between the center of STIM1 clusters and IP$_3$R1 or KRAP ($n$ = 398–803; one-way ANOVA, P < 0.0001).

μm away. Of note, for this comparison, SOCE microdomain was estimated from EM whereas STIM1–IP$_3$R1 distance from confocal images.

### STIM1 and PMCA

We next expressed the PM Ca$^{2+}$ATPase 4b (GFP-PMCA4b) and measured its localization relative to STIM1–Ch before and after store depletion. At rest, as expected, PMCA localizes to the PM and STIM1 to the ER resulting in no colocalization (Fig. 3, A and B, Ctr). Following store depletion, the PCC between STIM1 and PMCA4b increased slightly (Fig. 3 B), although the absolute co-localization remained extremely low compared with STIM1 and Orai1 colocalization (Fig. 3 B). The small increase in colocalization between STIM1 and PMCA is due to the PM translocation of STIM1 as indicated by the decreased axial distance between the two proteins following store depletion (Fig. 3 C). The intensity of the PMCA4b signal was not modified by the presence of the STIM1 cluster (Fig. 3, A and D). In the z-axis, both proteins displayed a similar profile, indicative of their close PM and ERPMCS localization (Fig. 3 E). Together, these analyses argue against any redistribution of PMCA4b in response to store depletion where it remains diffuse at the PM including within the SOCE microdomain.

### Engineering a Ca$^{2+}$ tunneling inhibitor (CaTAr)

Blocking Ca$^{2+}$ tunneling specifically without affecting SOCE or Ca$^{2+}$ release is conceptually difficult as tunneling requires both pathways. It is therefore not possible to block SOCE or IP$_3$Rs (Fig. S5 A). One could inhibit SERCA as it funnels Ca$^{2+}$ from SOCE to distant IP$_3$Rs to empower tunneling; however, a global SERCA block is not viable, as it would lead to store depletion and constitutive SOCE activation due to the inherent ER Ca$^{2+}$ leak. But would it be possible to inhibit only the SERCA pool that surrounds SOCE clusters at ERPMCS? Should this be possible it would produce a specific tunneling blocker without affecting SOCE or Ca$^{2+}$ release (Fig. 3 F).

We noticed from studies colocalizing the artificial ERPMCS marker MAPPER and STIM1, that, surprisingly, the two proteins do not colocalize after store depletion. At rest when stores are full MAPPER localizes to ERPMCS with some faint unclustered STIM1 visible at high magnification (Fig. 3 G, Ctr). Interestingly, following store depletion MAPPER and STIM1 localize to distinct subdomains laterally within ERPMCS (Fig. 3 G, CPA), where

MAPPER is clearly excluded from STIM1 clusters (Fig. 3 G), as previously reported (Carreras-Sureda et al., 2023; Henry et al., 2022). Time-lapse imaging of the evolution of STIM1 clustering at ERPMCS shows that STIM1 moving along microtubule tracks at rest, clusters in response to store depletion within the ERPMCS, and pushes out MAPPER to form a subdomain devoid of MAPPER within the same ERPMCS (Video 1). Orthogonal sections through the clusters confirm that after store depletion STIM1 and MAPPER are within the same axial plane but do not colocalize (Fig. 3 H). We measured the STIM1 to MAPPER nearest neighbor edge–edge distance, which was close to 0; while STIM1–STIM1 and MAPPER–MAPPER were far apart (Fig. S5 C). This argues that STIM1 clusters within a MAPPER-filled contact site.

We obtained a similar separation between STIM1 clusters and a shorter version of MAPPER (MAPPER-S) (Fig. S5, D and E), which brings the ER and PM closer together (within 10 nm) (Chang et al., 2013), as previously shown (Henry et al., 2022), arguing that the separation between MAPPER and STIM1 is not dependent on the gap distance between the ER and PM.

The distribution of MAPPER relative to STIM1 within ERPMCS argues that STIM1–Orai1 clusters form a privileged subdomain that prevents other molecules from colocalizing with it. To test this possibility, we assessed the localization of STIM1 relative to two endogenous ER–PM tethers, E-Syt2 and TMEM24 (Chang et al., 2013; Chen et al., 2019). In contrast to MAPPER, both E-Syt2 and TMEM24 colocalize with STIM1 clusters, as assessed at the whole cell level (Fig. S6 A), by quantifying colocalization using PCC (Fig. S6 B), and at high magnification within STIM1 clusters using line scan across individual clusters (Fig. S6 C).

The distinct distribution of MAPPER following store depletion raised the intriguing possibility that it could localize close to or within SERCA-rich cortical ER subdomains, especially since MAPPER has been shown to extend ERPMCS (Carreras-Sureda et al., 2023; Henry et al., 2022). Should this be the case it would provide an ideally targeted molecule to inhibit cortical SERCA. But how to specifically inhibit SERCA using MAPPER? For this, we used the transmembrane domain of two well-characterized SERCA inhibitors in muscle cells, phospholamban (PLN) and sarcolipin (SLN), which importantly modulate SERCA function through their transmembrane (TM) domains (Shaikh et al., 2016). Therefore, to create a cortical SERCA inhibitor—which

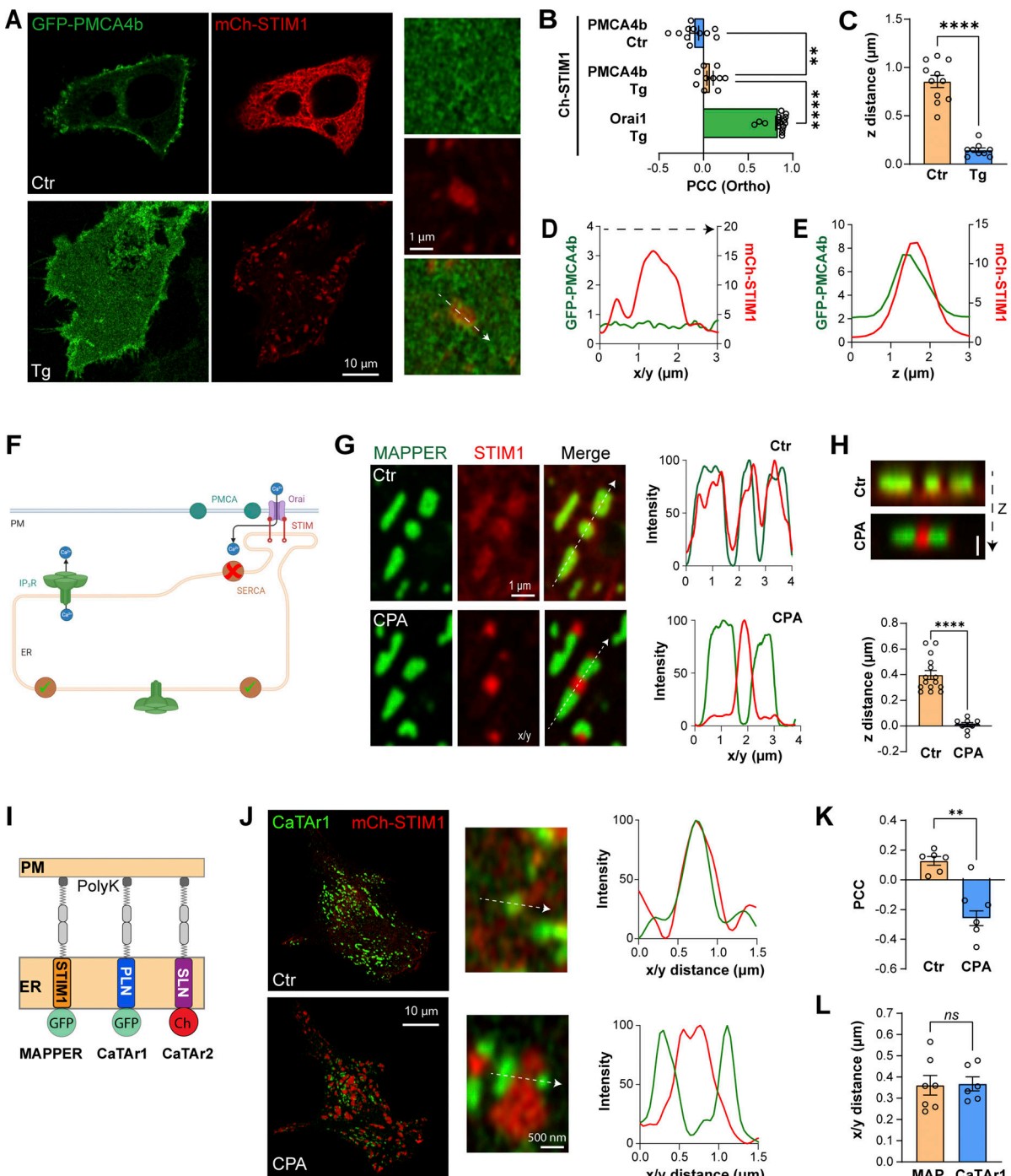

Figure 3. **Localization of PMCA4b and design of a tunneling inhibitor. (A)** Distribution of GFP-PMCA4b and mCh-STIM1 in HeLa cells before (Ctr) and after (Tg) store depletion. The control confocal image is taken in the middle of the cell and images after store depletion at the PM plane. **(B)** Bar chart summarizing the PCC between mCh-STIM1, GFP-Orai1 (used as a maximum reference), and GFP-PMCA4b, measured in the orthogonal plane (n = 10–16; one-way ANOVA, P < 0.0001). **(C)** Quantification of the Z distance between mCh-STIM1 and GFP-PMCA4b peaks (n = 9–11; unpaired t test, P < 0.0001). **(D and E)** Intensity profiles along the x/y (D) and z (E) axis of mCh-STIM1 and GFP-PMCA4b across a STIM1 cluster (as indicated in A). **(F)** Cartoon summarizing the strategy to block cortical SERCA to specifically inhibit tunneling. **(G)** High magnification Airyscan images of GFP-MAPPER and mCh-STIM1 at rest (Ctr) and after store depletion (CPA). Line scans as indicated on the merged image show the colocalization of MAPPER (green) with the diffuse unclustered STIM1 (red) at rest, and the separation of clustered STIM1 from MAPPER. **(H)** Orthogonal sections of GFP-MAPPER and mCh-STIM1 before (Ctr) and after store depletion (CPA). The bar chart reports the distance in the z-axis between maximum intensities of GFP-MAPPER and mCh-STIM1 signals before (Ctr) and after store depletion (CPA) (n = 8–15; unpaired t test; P < 0.0001). **(I)** Cartoon depicting the structure of the Ca²⁺ tunneling attenuators (CaTAr1 and 2) compared to MAPPER. **(J)** Confocal images at the PM plane of CaTAr1 and mCh-STIM1 before (Ctr) and after (CPA) store depletion. The intensity plots using the white arrows in the high-magnification images report the colocalization of STIM1 and CaTAr1 at rest and their separation after store depletion. **(K)** Bar chart summarizing the PCC between CaTAr1 and mCh-STIM1 before (Ctr) and after store depletion (CPA) (n = 6; paired t test, P = 0.002). **(L)** Bar chart summarizing the distance between STIM1 clusters and CaTAr1 or MAPPER (MAP) (n = 6–7; unpaired t test, P = 0.91). Cartoons were generated using Biorender.

is a specific tunneling blocker—we replaced the MAPPER TM domain with either the TM domain of PLN or SLN (Fig. 3 I). This generated GFP–PLN–MAPPER, which we named CaTAr1 for Ca²⁺ Tunneling Attenuator 1, and Ch–SLN–MAPPER named CaTAr2. When coexpressed with mCh-STIM1 or STIM1-CFP, both CaTAr1 (Fig. 3 J) and CaTAr2 (Fig. S7) localized adjacent to clustered STIM1 but were excluded from STIM1 clusters. They segregate apart from STIM1 clusters following store depletion (Fig. 3, J–L and Fig. S7 A; and Video 2). CaTAr2 also colocalizes with MAPPER (Fig. S7 B), and was isolated from cortical IP₃R1–GFP (Fig. S7 C). Together, these results argue that the CaTAr constructs localize around but not within STIM1 clusters.

To confirm that PLN inhibits SERCA in HeLa cells, we expressed wild-type PLN and showed that it distributes throughout the entire ER, where it colocalizes with STIM1 (Fig. S8 A) and leads to store depletion with a significant reduction of histamine-induced Ca²⁺ release in PLN expressing cells (Fig. S8, B–D), consistent with global SERCA inhibition. To further directly show that CaTAr inhibits SERCA, we engineered a CaTAr1 version lacking the terminal polylysine domain that targets it to ERPMCS (CaTArΔpolyK). CaTArΔpolyK localizes diffusively to the ER (Fig. S8 E), in contrast to the punctate ERPMCS localization of CaTAr1. Histamine induces a robust Ca²⁺ release signal from control cells but not from cells in the same dish that express CaTArΔpolyK (Fig. S8, F–H). These data show that CaTAr inhibits SERCA.

## CaTAr inhibits Ca²⁺ tunneling
We next asked whether CaTAr blocks Ca²⁺ tunneling and whether it affects SOCE or Ca²⁺ release. We first tested whether the CaTAr backbone, i.e., MAPPER, affects tunneling. Applying our standard tunneling protocol to HeLa cells expressing MAPPER–GFP and loaded with the Ca²⁺ indicator Calbryte590 leads to similar Ca²⁺ tunneling amplitudes in control and MAPPER expressing cells (Fig. 4, A–C), showing that MAPPER expression does not significantly affect Ca²⁺ tunneling amplitude. However, the rising phase of the tunneling signal was slower in MAPPER-expressing cells (Fig. 4 D), arguing that somehow MAPPER modulates the rate of tunneling potentially by altering ERPMCS structure.

We then tested the effect of CaTAr1 expression on agonist-dependent Ca²⁺ release and SOCE. CaTAr1 expression did not alter the levels of Ca²⁺ release in response to histamine in Ca²⁺-free media (Fig. 4, E and F), showing that it does not modulate agonist-dependent Ca²⁺ release. Similarly, CPA-dependent Ca²⁺ release was not affected by CaTAr1 expression arguing against any modulation of ER Ca²⁺ leak or Ca²⁺ store content (Fig. 4 E). We then measured SOCE after store depletion with CPA and a wash period to allow SERCA to be active. SOCE levels were not affected by CaTAr1 expression (Fig. 4, G and H). However, as this SOCE signal was quite small, modest effects on SOCE due to CaTAr1 expression may be difficult to quantify. To further test the effect of CaTAr1 expression on SOCE, we employed both thapsigargin-induced SOCE and NFAT nuclear translocation as a functional reporter of SOCE levels within ERPMCS. CatAr1 had a mild inhibitory effect on SOCE that was induced by thapsigargin (Fig. 4 I), potentially due to modulation of ERPMCS as similar inhibitory effects have been documented with MAPPER (Henry et al., 2022). In contrast, CaTAr1 expression did

not affect NFAT nuclear translocation (Fig. 4 J), which is the more physiological SOCE reporter.

Ca²⁺ tunneling in contrast was significantly and dose-dependently inhibited by CaTAr1 expression (Fig. 4, K–M and Fig. S9 A). At the individual cell level, the extent of inhibition of Ca²⁺ tunneling correlates with the expression levels of CaTAr1 (Fig. 4, K–L), with complete inhibition in cells with high CaTAr1 expression (Fig. 4, K–L, cell#3). At the population level, the amplitude of the tunneling signal was significantly reduced (by ~47%) in cells expressing CaTAr1 compared to those with no expression in the same dish (Fig. 4 M).

A second application of histamine following Ca²⁺ tunneling confirms the ability of the CatAr1-expressing cells to refill their stores (Fig. 4 L). This release after tunneling and store refilling shows a small reduction in amplitude in cells expressing CaTAr1 (Fig. 4 N). Interestingly, as discussed above, we did not detect such a reduction in His-induced Ca²⁺ release in Ca²⁺-free media, which was similar in control and CatAr1 expressing cells (Fig. 4, E and F). In Ca²⁺-free media, the Ca²⁺ signal depends solely on Ca²⁺ release from stores as there is no influx. As we observed a decrease in the Ca²⁺ release signal in response to histamine in Ca²⁺-containing but not Ca²⁺-free media, it indicates that a fraction of the Ca²⁺ during the release phase in response to agonist is through tunneling. This is important as it argues that tunneling is activated early on following Ca²⁺ release due to partial store depletion while IP₃ levels remain high.

Similar results were obtained with CaTAr2, which inhibited Ca²⁺ tunneling with no effect on Ca²⁺ release (Fig. S7, D–F). CaTAr2 was less potent than CaTAr1 with ~24% reduction in Ca²⁺ tunneling in CaTAr2 expressing cells as compared with non-expressing cells in the same dish (Fig. S7 E).

Collectively, these results show that the CaTAr constructs are specific tunneling inhibitors that reduce Ca²⁺ tunneling without substantially affecting either Ca²⁺ release from stores or SOCE. This provides a powerful tool to test the role and contribution of tunneling to cellular and physiological responses.

## Ca²⁺ tunneling is independent of PLC activation
For the studies conducted so far, we've induced tunneling using GPCR-coupled agonists that would activate phospholipase (PLC) leading to the production of IP₃ and DAG. We therefore wanted to rule out any contribution from DAG or other branching pathways downstream of PLC to Ca²⁺ tunneling. We performed the tunneling protocol using caged IP₃ instead of an agonist (Fig. 5). Ca²⁺ stores were depleted in a Ca²⁺-free medium with CPA followed by a washout period and then Ca²⁺ was added to induce SOCE (Fig. 5, A and B). As soon as SOCE began to develop, we uncaged IP₃ using a UV pulse which was previously calibrated in the same batch of cells to induce Ca²⁺ release (Fig. 5, A and B). Summary time courses from multiple experiments are shown in Fig. 5 A, while Fig. 5 B shows a representative experiment with an expanded time course around the uncaging pulse. IP₃ uncaging results in a large increase in the cytosolic Ca²⁺ transient (Fig. 5, A–C), showing that IP₃ alone is sufficient to induce Ca²⁺ tunneling downstream of SOCE. IP₃ uncaging led to an ~6.5-fold increase in cytosolic Ca²⁺ compared

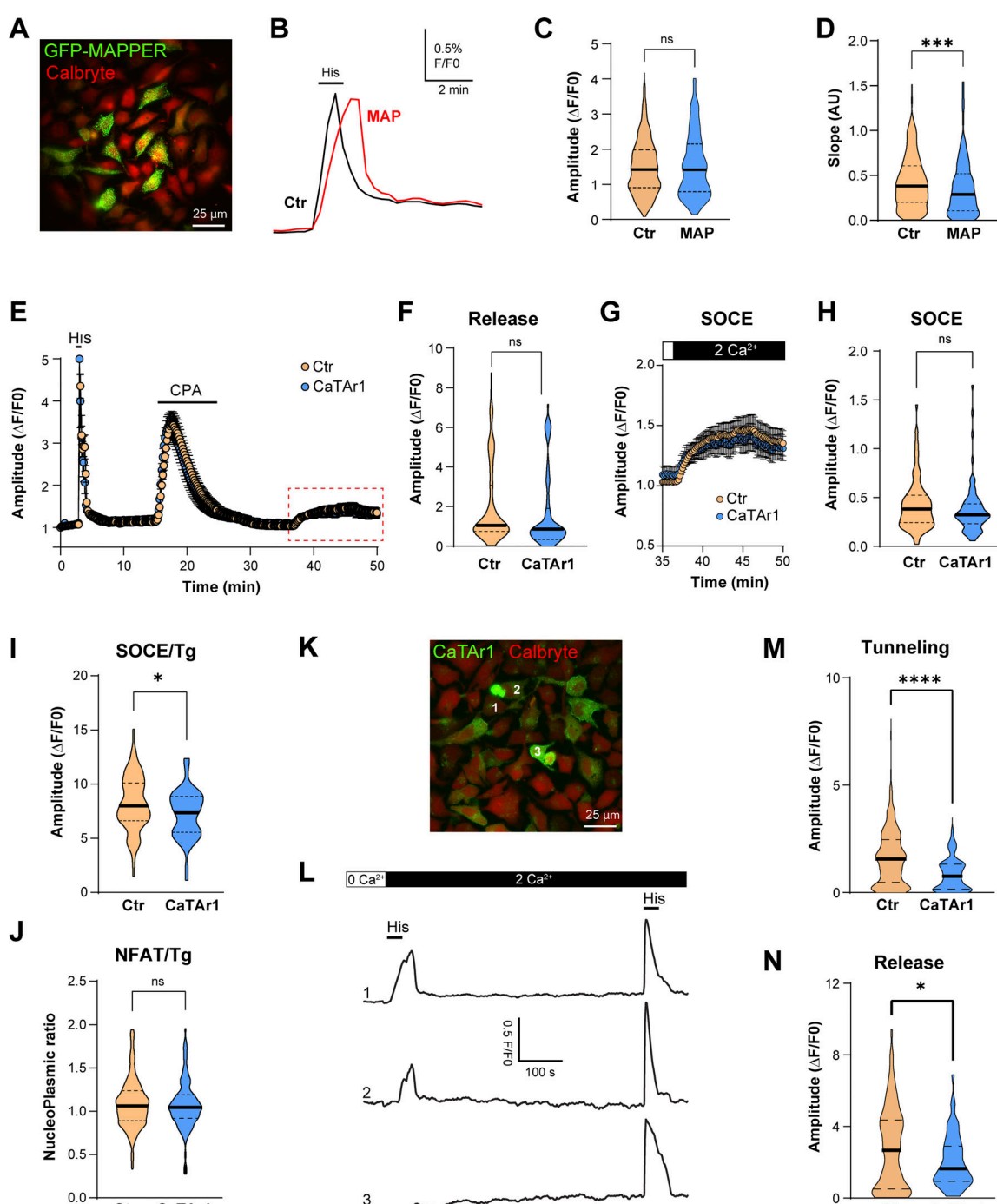

Figure 4. **CaTAr1 inhibits Ca²⁺ tunneling. (A)** HeLa cells expressing GFP-MAPPER and loaded with the Ca²⁺ indicator Calbryte 590. **(B)** Changes in intracellular Ca²⁺ during a tunneling experiment from an untransfected cell (Ctr) and a cell expressing GFP-MAPPER (MAP). **(C)** Violin plots of the amplitude of the tunneling signal in Ctr and MAPPER expressing cells (MAP) ($n$ = 439/118; 12 dishes; unpaired $t$ test, P = 0.65). **(D)** Violin plot of the slope of the initial phase of tunneling in MAPPER expressing cells $n$ = 427/110, unpaired $t$ test, P = 0.0008). **(E)** Example averaged traces of Ca²⁺ release and SOCE in control (Ctr; no detectable CaTAr1 expression) and CatAr1 expressing cells from the same dish. Histamine-induced (His, 100 μM) Ca²⁺ release from stores was followed by CPA to deplete Ca²⁺ store, a wash period in Ca²⁺-free media to allow for SERCA to be active, and then the addition of Ca²⁺ to activate SOCE ($n$ = 9/5). **(F)** Violin plots summarizing the levels of Ca²⁺ release in response to histamine in a Ca²⁺-free solution ($n$ = 192/77; 5 dishes; unpaired $t$ test, P = 0.085). **(G and H)** Enlarged traces from the red rectangle in E and violin plots summarizing the levels of SOCE in cells that did not express CatAr1 (Ctr) and CaTAr1 expressing cells ($n$ = 192/77, unpaired $t$ test, P = 0.10). **(I)** SOCE amplitude after thapsigargin application (1 μM) on control cells (Ctr) and cells expressing CaTAr1 ($n$ = 125/51, 3 dishes, unpaired $t$ test, P = 0.0175). **(J)** Quantification of NFAT nuclear translocation in Ctr and CaTAr1 expressing cells ($n$ = 269/109, 5 dishes, unpaired $t$ test, P = 0.58). **(K)** HeLa cells expressing CaTAr1 and loaded with Calbryte 590. The numbers correspond to the cells in L. **(L)** Traces showing the Ca²⁺ tunneling transient in response to His+Ca²⁺ following the standard store depletion with CPA and wash (not shown for clarity). A second histamine application after a delay in Ca²⁺-containing media confirms that all cells including those expressing CatAr1 refill their stores. **(M and N)** Violin plots summarizing the levels of Ca²⁺ tunneling (M, $n$ = 262/83, six dishes, unpaired $t$ test, P < 0.0001) and Ca²⁺ release in response to His following the second application in Ca²⁺-containing media (N, $n$ = 182/56, five dishes, unpaired $t$ test, P = 0.0250).

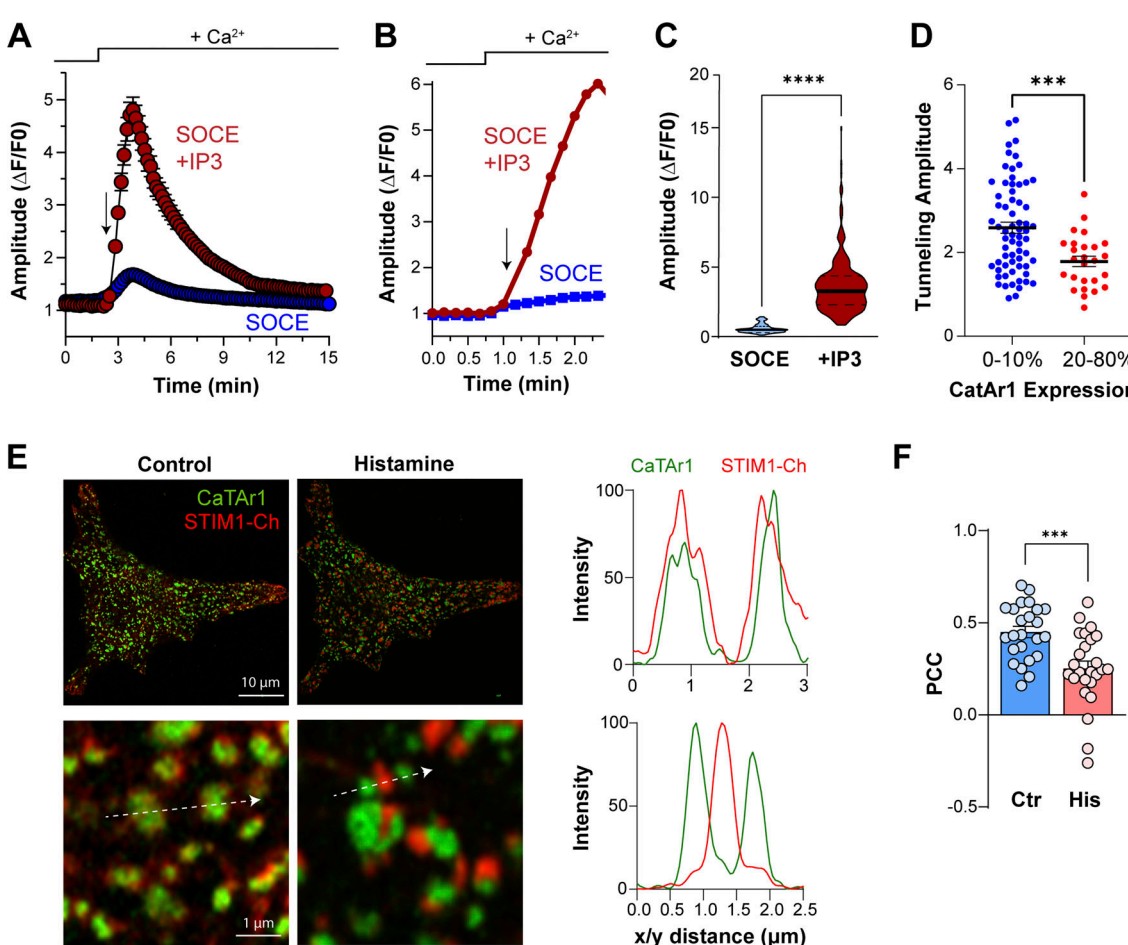

Figure 5. **Uncaging of IP₃ and histamine-induced clustering. (A)** HeLa cells were loaded with caged-IP₃ together with Calbryte 590. Ca²⁺ stores were depleted using CPA and the CPA was removed by a 20 min wash. SOCE was triggered by the re-addition of Ca²⁺. After 20 s a UV flash was triggered to induce IP₃ release (arrow). Average example traces are represented with uncaging of IP₃ (red circles, from 97 cells) and without (blue circles, from 29 cells). **(B)** Single traces illustrating the change in the Ca²⁺ signal during uncaging (arrow, red trace) compared with SOCE only (blue trace) over an expanded time course around the uncaging pulse. **(C)** Bar chart summarizing the signal amplitude during SOCE and when tunneling is induced by IP₃ uncaging (*n* = 274/105, P < 0.0001). **(D)** Pilot experiment showing inhibition of IP₃-induced tunneling by CatAr1 expression. The relative normalized expression levels of CatAr1 are indicated on the x-axis (*n* = 66/26, P = 0.0007). **(E)** Airyscan images of HeLa cells expressing STIM1-Ch and CaTAr1 before and after application of histamine (100 μM). **(F)** Intensities obtained from virtual line scans in the x/y plane (arrows) across CaTAr1 (green) and STIM1 (red) signals before and after the application of histamine. Both signals are colocalized at rest but the CaTAr1 segregates away from STIM1 clusters following histamine application. **(F)** Bar chart summarizing the colocalization intensity between CaTAr1 and STIM1 before (Ctr) and during histamine application (His) (*n* = 24, paired-*t* test, P = 0.0009).

with SOCE alone (Fig. 5 C). Furthermore, we validated in a pilot experiment the inhibition of tunneling induced by IP₃ uncaging by CatAr1 expression (Fig. 5 D).

As we wanted to use CatAr1 to block physiological tunneling, we then tested its localization relative to STIM1 clusters in response to the agonist. Histamine led to STIM1 clustering at ERPMCS and the separation of CatAr1 to the periphery of STIM1 clusters (Fig. 5 E) in a similar fashion to CPA (Fig. 3 G). This was associated with reduced colocalization between STIM1 and CatAr1 in response to histamine (Fig. 5 F).

### CaTAr inhibits Cl⁻ secretion in human sweat cells

Sweating depends on SOCE as patients with reduced SOCE due to mutations in either STIM1 or Orai1 suffer from anhidrosis (McCarl et al., 2009). The contributions of SOCE and the Ca²⁺-activated Cl channel ANO1 to sweat production have been demonstrated in situ in mice using genetic or chemical

inhibition of SOCE, and in the immortalized human eccrine sweat gland cell line NCL-SG3, respectively (Concepcion et al., 2016; Lee and Dessi, 1989). Therefore, NCL cells represent a good model to test the role of tunneling in sweating and the effectiveness of CaTAr. We previously showed that ANO1 segregates away from SOCE clusters in response to store depletion in *Xenopus* oocytes, thus requiring tunneling to deliver Ca²⁺ entering through SOCE to ANO1 (Courjaret and Machaca, 2014).

We localized ANO1 relative to STIM1 in NCL cells using either an antibody against ANO1 or by expressing ANO1–GFP together with Ch–STIM1. ANO1 localizes to the PM and STIM1 to the ER at rest (Fig. 6 A; and Fig. S9, B and C). Following store depletion, ANO1 was largely excluded from SOCE clusters and localized at their periphery (Fig. 6 A; and Fig. S9, B and C). We quantified ANO1–STIM1 colocalization using PCC as compared with STIM1–Orai1. Orai1 colocalizes with STIM1 in NCL cells to a similar level as in HeLa cells (Fig. 6 B). In contrast, ANO1

(endogenous or GFP-tagged) was largely isolated from STIM1 clusters (Fig. 6 B). This separation was further illustrated by the STIM1 to ANO1 peak-to-peak distance in the lateral dimension (x/y) (Fig. 6 C), while in the axial dimension STIM1, Orai1, and ANO1 localized to the same Airyscan focal plane (Fig. 6 D).

To induce Cl⁻ secretion in NCL cells, we used trypsin to activate the proteinase-activated receptor 2 (PAR2), which led to $Ca^{2+}$ release (Fig. 6 E). Trypsin induces a rise in cytosolic $Ca^{2+}$ that was reduced in amplitude (18.8 + 0.9%), and in a more pronounced fashion in its duration (48% + 0.8%) by the CRAC channel inhibitor BTP2 (Fig. 6 E). This argues for a role for SOCE/tunneling in generating the maximal $Ca^{2+}$ signal in response to agonist stimulation. We then expressed in NCL cells the membrane-bound Cl⁻ sensor mbYFPQS (Watts et al., 2012), which localizes diffusely to the PM, including to microvilli away from the STIM1 clusters (Fig. 6 F). Orthogonal sections through confocal z-stacks confirmed the membrane localization of the sensor (Fig. 6 F). We subjected cells coexpressing mbYFPQS and STIM1 to our standard tunneling protocol and used a line scan across a SOCE cluster and the cell margin to follow the temporal progression of Cl⁻ secretion using TIRF microscopy (Fig. 6 G). The kymographs showed a rise in mbYFPQS fluorescence (indicative of Cl⁻ secretion) far from the STIM1 cluster (the site of $Ca^{2+}$ entry) (Fig. 6 G). This supports $Ca^{2+}$ tunneling from the SOCE point source entry to the distal ANO1 channels (Fig. 6 G). Note that the intensity of the STIM1 cluster decreased along the kymograph following store refilling (Fig. 6 G), which is indicative of clustered STIM1 dissociation. These data show that $Ca^{2+}$ tunneling supports Cl secretion distally to the SOCE clusters.

To assess the contribution of $Ca^{2+}$ tunneling to Cl⁻ secretion, we used CaTAr2 (mCh-SLN-MAPPER) to avoid overlap with the mbYFPQS fluorescence. ANO1 localizes around CaTAr2 clusters and significantly further away at PM extensions, and this localization was not affected by store depletion (Fig. S10). Expression of CaTAr2 led to a slower and smaller Cl⁻ secretion signal (Fig. 6 H). Quantification of Cl⁻ secretion from mbYFPQS fluorescence at 3 min post-trypsin shows a significant decrease in CaTAr2-expressing cells (Fig. 6 I). Collectively these results show that tunneling is an important contributor to the activation of the $Ca^{2+}$-activated Cl⁻ channels in NCL sweat gland cells.

### Tunneling supports sweating *in vivo*

We next wanted to evaluate the potential contribution of tunneling *in vivo*. We used the paw sweat test as a model since it has been successfully used in the past as an indicator of SOCE's contribution to sweating in mice (Concepcion et al., 2016; Yu et al., 2022). We used an adenoviral vector to express CaTAr1 under the control of the CMV promoter in the paw of mice. CaTAr1 expression (GFP) could be detected 4 days following adenovirus injection (Fig. 7 A). We injected an adenovirus expressing GFP alone as a control. Sweating was evaluated by applying starch iodine to the paw and measuring the area covered by the black dots (Fig. 7, B and C). We observe a significant reduction in sweating in animals expressing CaTAr1 as compared with GFP (Fig. 7, B–D). This supports the reduction in Cl⁻

secretion following tunneling inhibition in NCL cells and shows that tunneling is an important contributor to sweat production in vivo.

## Discussion

Various agonists generate intracellular $Ca^{2+}$ signals through the activation of PLC-coupled receptors to produce $IP_3$ that releases $Ca^{2+}$ from ER stores. The release phase is followed by $Ca^{2+}$ influx from the extracellular space through SOCE. PLC-generated lipid messengers can also activate members of the TRP family, some of which are $Ca^{2+}$-permeant (Clapham et al., 2001; Venkatachalam and Montell, 2007). However, SOCE is highly $Ca^{2+}$ selective and is responsible for the prolonged low amplitude $Ca^{2+}$ signal following store depletion. The SOCE signal inactivates when $IP_3$ levels drop, leading to termination of $Ca^{2+}$ release through $IP_3Rs$, store refilling by SERCA, and dissociation of STIM1–Orai1 clusters. The work presented here shows that agonist-mediated $Ca^{2+}$ signaling has a third central component, $Ca^{2+}$ tunneling (Fig. 7 E).

We previously showed that $Ca^{2+}$ tunneling is primarily cortical and have argued that this is because of $IP_3R$ distribution and its large $Ca^{2+}$ conductance compared with Orai1 (Courjaret et al., 2018; Taylor and Machaca, 2019). The geography of the primary $Ca^{2+}$ tunneling effectors (SERCA, PMCA, and $IP_3R1$) relative to STIM1 outlined here supports and extends this conclusion. We localized endogenous STIM1, SERCA, and $IP_3R1$ to avoid issues with overexpressed tagged proteins. We show that a population of licensed $IP_3R1$ colocalize with KRAP cortically close to the PM, consistent with previous studies (Taylor and Machaca, 2019; Thillaiappan et al., 2017, 2021). On average, licensed $IP_3R1s$ localize ~1 μm away from a STIM1 cluster and can be up to ~2 μm away. This effectively expands the spatial reach of SOCE to cortical effectors that are 1–2 μm away. $Ca^{2+}$ that enters the cell within the SOCE microdomain and is taken up into the ER by SERCA is released by the nearest open $IP_3R$ it encounters. Given the large conductance of $IP_3R$, the limiting factors for this $Ca^{2+}$ tunneling would be the combined rate of $Ca^{2+}$ entry through Orai1 and SERCA uptake. The SOCE $Ca^{2+}$ transient would be mostly limited to the SOCE microdomain (100–300 nm in diameter) and thus would activate effectors within this microdomain. Therefore, the localization of endogenous $IP_3R1$ cortically - close to the PM axially but distal to the SOCE microdomain laterally - is well-suited to support tunneling by expanding the spatial extent of SOCE (up to 10-fold) to reach distal $Ca^{2+}$ dependent effectors. We show that this is indeed the case as $Ca^{2+}$ tunneling activates larger and faster $Ca^{2+}$ signals and associated Cl⁻ currents in primary salivary cells (Fig. 1, D and E). We further show that blocking tunneling in sweat cells (Fig. 6, H and I) or in vivo in mice (Fig. 7, A–D) inhibits Cl⁻ secretion and sweating, respectively.

Previous imaging studies in different cell types, including our own, used tagged overexpressed SERCA and argued for its close association with STIM1 (Alonso et al., 2012; Courjaret and Machaca, 2014; Manjarrés et al., 2010; Sampieri et al., 2009; Vaca, 2010). In some studies, this was supplemented by FRET and co-IP experiments (Jha et al., 2019; Sampieri et al., 2009). We were concerned that overexpression or the tag itself could interfere with localization, especially when using high-resolution

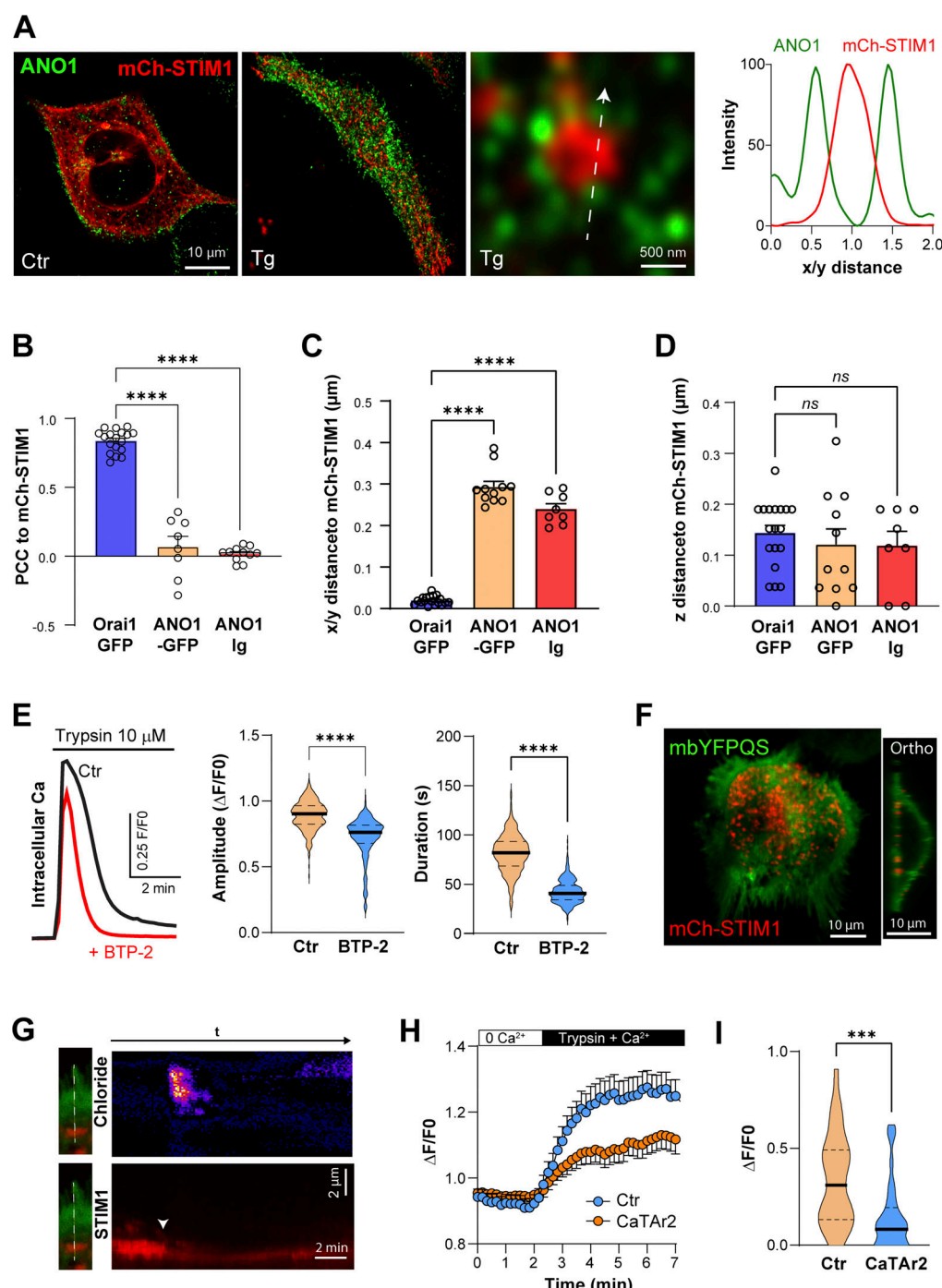

Figure 6. **Tunneling in NCL-SG3 cells. (A)** Localization of ANO1 by immunocytochemistry and mCh-STIM1 before (Ctr) and after store depletion (Tg). An intensity plot performed along a line (arrow) crossing a STIM1 cluster illustrates the separation of STIM1 and ANO1 at the PM focal plane. **(B–D)** Comparative localization at the PM plane of mCh-STIM1 with ANO1–GFP and the endogenous ANO1 protein (ANO1–Ig) following store depletion. As indicated by the PCC and the peak-to-peak distances the two proteins do not colocalize laterally but are localized at the same optical plane (z distance) (n = 8–18; one-way ANOVA, P < 0.0001 for B and C, P = 0.7 for D). **(E)** Intracellular Ca²⁺ elevation induced by trypsin application in NCL-SG3 cells loaded with Fluo4-AM. The application of the SOCE inhibitor BTP-2 (10 µM) reduces the amplitude and the duration of Ca²⁺ release (n = 379/402; unpaired t test, P < 0.0001). **(F)** Confocal images of NCL-SG3 cells expressing the Cl⁻ sensor mbYFPQS and mCh-STIM1 after store depletion. The orthogonal section through the cell indicates the PM localization of the chloride sensor. **(G)** Kymographs were measured during a tunneling event on cells expressing mbYFPQS and mCh-STIM1. The line passes through a SOCE cluster and an adjacent cell appendage labeled by mbYFPQS. The changes in Cl⁻ concentration (upper) are distal from the STIM1 cluster, which indicates the Ca²⁺ entry point. **(H)** Time course of the amplitude of the Cl⁻ signal induced by Ca²⁺ tunneling from NCL-SG3 cells in the same dish that either do not show any CaTAr2 expression (Ctr) and cells expressing CaTAr2. **(I)** Violin plots summarizing the amplitude of the Cl⁻ signal 3 min after trypsin and Ca²⁺ addition to stimulate tunneling (n = 32–33; unpaired t test, P = 0.0004).

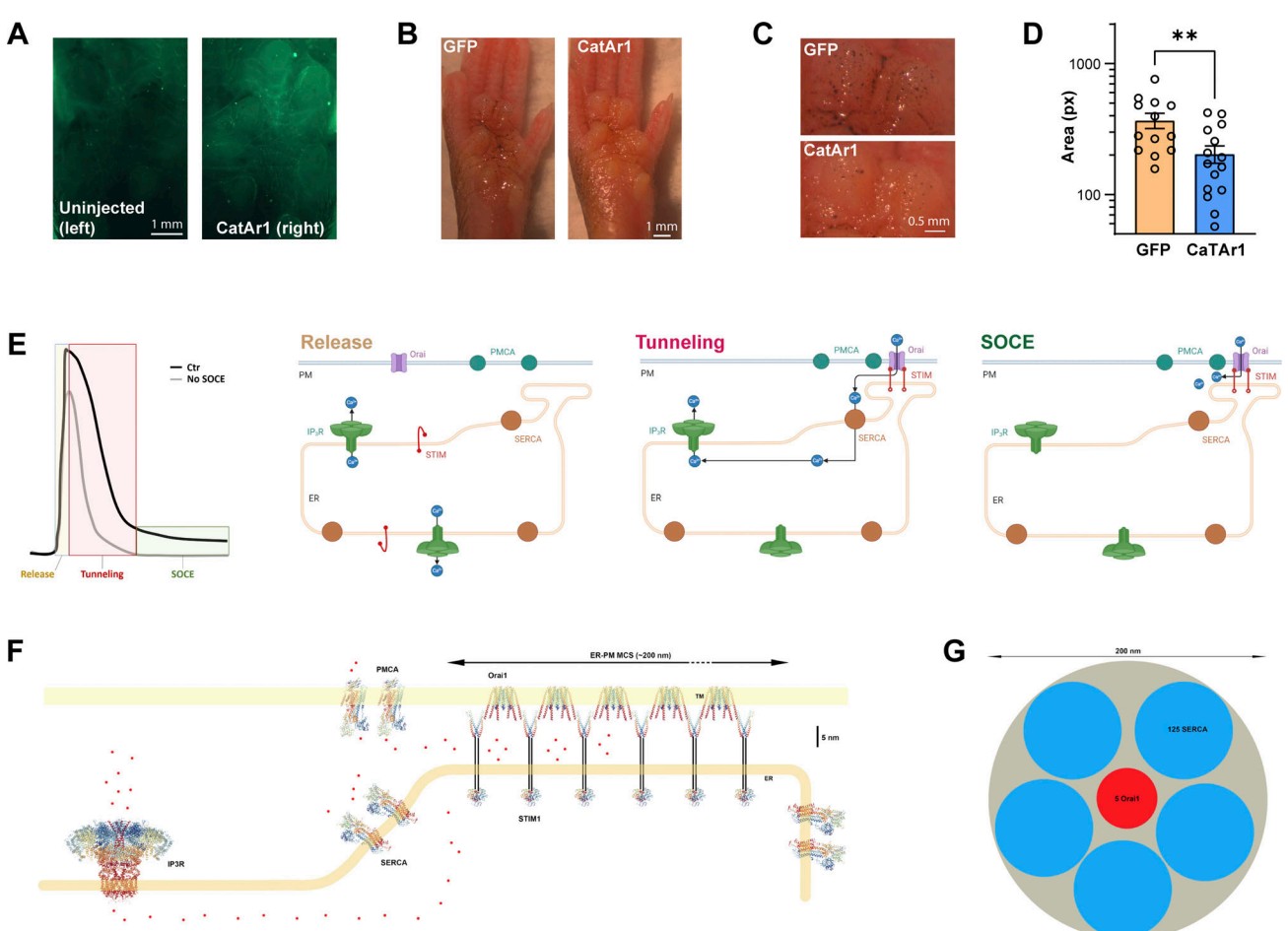

Figure 7. **Inhibition of sweating by CaTAr1. (A)** GFP fluorescence in the mouse paw injected with CaTAr1 (right) compared with the contralateral paw (left) on the same animal. **(B and C)** Sweating was visualized using the iodine/starch technique in GFP- and in CatAr1-expressing paws. **(D)** Bar chart summarizing the sweating areas recorded after 15 min in control (GFP) and CatAr1 expressing paws ($n$ = 13–15; unpaired $t$ test, P = 0.0074). **(E)** Cartoons illustrating the different phases of a typical agonist-driven $Ca^{2+}$ release signal with (Ctr) or without SOCE. The activation state and localization of the $Ca^{2+}$ effectors are shown for the different phases. **(F)** To scale illustration of the $Ca^{2+}$ effectors involved in tunneling. For STIM1 only the domains with the atomic structure solved are shown. **(G)** To scale depiction of 5 Orai1 channels and clusters of 125 SERCAs each.

microscopy, as we observed a differential localization of tagged (Fig. S4, F–L) and endogenous IP₃R1 (Fig. 2, K–N), where tagged IP₃R1 localized closer to STIM1 clusters. In contrast, we did not observe a differential localization between tagged and endogenous STIM1. Localization of endogenous SERCA2b shows that it is excluded from the SOCE microdomain, and rather surrounds it within a cortical ER domain that localizes close to the ERPMCS. This is consistent with functional studies arguing for SERCA being close to SOCE microdomains (Courjaret et al., 2018; Jousset et al., 2007). The localization of SERCA suggests a funnel around the SOCE microdomain that collects $Ca^{2+}$ ions that spillover out of ERPMCS and tunnel them into the ER to support IP₃R-mediated release at distal sites. Such a distribution would fit well with the slow transport rate for SERCA (~40 $Ca^{2+}$/s at $V_{max}$ [Hogan, 2015; Lytton et al., 1992]), requiring around 125 SERCA molecules to take up the $Ca^{2+}$ flowing through a single Orai1 channel (~5,000 ions/s [Hogan, 2015]). As SERCA localizes outside the SOCE microdomain, it would be expected to transport only a fraction of $Ca^{2+}$ ions that spill out of the SOCE microdomain. Is it possible to fit such a large number of SERCAs around a STIM1–Orai1 cluster?

Fig. 7 F shows a to scale rendition of the tunneling effectors based on published structures. Assuming 5 Orai1 channels within the SOCE microdomain and 125 SERCA/Orai1, the SERCAs required would fit within a 200 nm diameter ER domain (Fig. 7 G).

The gap between the ER and PM has been shown to be modulated by E-Syt isoforms (Fernández-Busnadiego et al., 2015). When the ERPMCS gap was artificially shortened using chemically induced linkers to below 6 nm, it prevented the stabilization of STIM1–Orai1 clusters and favored their enrichment at the junction periphery (Várnai et al., 2007). Furthermore, similar to our findings, Henry et al. showed that MAPPER localizes to a different subdomain than STIM1 within ERPMCS, whereas a longer tether Sec22 colocalized with STIM1 (Henry et al., 2022), similar to what we show herein for two endogenous tethers TMEM24 and E-Syt2 (Fig. S6). Based on these findings Henry et al. argued that STIM1–Orai1 clusters localize to the periphery of ERPMCS. However, our findings regarding the localization of SERCA and that of CaTAr (as an inhibitor of cortical junctional SERCAs) suggest that the STIM1–Orai1 clusters localize to the center of ERPMCS. We took advantage of this

unusual MAPPER localization at the periphery of STIM1 clusters to develop a specific tunneling inhibitor (CaTAr). We replaced the MAPPER TM domain with either PLN or SLN to specifically block the subpopulation of cortical SERCAs close to SOCE microdomains. This effectively reduced tunneling without substantially affecting $Ca^{2+}$ release or SOCE (Fig. 4).

CaTAr-mediated inhibition of tunneling lowers $Cl^-$ secretion in NCL sweat cells (Fig. 6, H and I), and more importantly, it reduces sweating *in vivo* in mice injected with a virus expressing CaTAr (Fig. 7, A–D). Furthermore, we show that tunneling activates larger $Cl^-$ currents in primary salivary cells (Fig. 1). This is consistent with an early study on HSG cells from the salivary gland showing that ER $Ca^{2+}$ and $IP_3$-dependent $Ca^{2+}$ release are needed for optimal activation of $Ca^{2+}$-activated $K^+$ currents (Liu et al., 1998), a mechanism similar to $Ca^{2+}$ tunneling. In addition, $Ca^{2+}$ tunneling has been implicated in secretion in the exocrine pancreas (Mogami et al., 1997; Petersen et al., 2017; Petersen and Tepikin, 2008). Collectively, these findings argue for an important role for $Ca^{2+}$ tunneling in fluid secretion in exocrine glands by modulating $Cl^-$ and $K^+$ channels to support vectorial ion and fluid transport.

Based on both the architecture of the $Ca^{2+}$ signaling machinery and the functional data following tunneling inhibition, we propose three phases in the $Ca^{2+}$ signal following agonist stimulation (Fig. 7 E). At rest with $Ca^{2+}$ stores full, $IP_3$ releases $Ca^{2+}$ from stores resulting in the initial cytosolic $Ca^{2+}$ rise (Fig. 7 E, Release). This leads to store depletion and SOCE activation. During the early phases of store depletion, $IP_3Rs$ would still be open due to the presence of $IP_3$ from receptor activation. These conditions would support $Ca^{2+}$ tunneling as SOCE is active and $IP_3Rs$ are open (Fig. 7 E, Tunneling). When $IP_3$ levels fall $IP_3Rs$ close, leading to the termination of tunneling and a smaller global $Ca^{2+}$ signal as $Ca^{2+}$ entry would localize primarily to SOCE microdomains (Fig. 7 E, SOCE). Functional evidence supports this model. In salivary cells, $Ca^{2+}$ signals and $Cl^-$ currents are larger during tunneling as compared with SOCE alone (Fig. 1, D and E). They are also induced significantly faster (Fig. S1, A–C), arguing that tunneling enhances the speed of the response by delivering $Ca^{2+}$ more efficiently to its target, in this case, ANO1. In sweat cells and *in vivo* in the sweat gland, blocking tunneling using CaTAr inhibits $Cl^-$ secretion (Fig. 6, H and I) and sweat production (Fig. 7, A–D).

In the model proposed in Fig. 7 E, tunneling is predicted to modulate the duration of the agonist-induced $Ca^{2+}$ rise, a conclusion supported by the signal observed when SOCE is blocked (Fig. 7 E, No SOCE). This modulation would depend on multiple factors: (1) the rate of $IP_3$ production and degradation as tunneling depends on high $IP_3$ levels; (2) the density of ERPMCS as sites for SOCE; (3) the density of licensed $IP_3Rs$; (4) the levels of STIM1, Orai1, PMCA, and SERCA; and (5) the distances between SOCE clusters, $IP_3Rs$, and distal effectors. The dependency on these multiple factors allows cells the flexibility to modulate the extent of tunneling to fit their physiological needs. Despite the limited cell types tested to date, we have some confirmation of these predictions. For example, $Ca^{2+}$ tunneling in frog oocytes induces a 30-fold larger $Cl^-$ current compared with SOCE alone (Courjaret and Machaca, 2014), whereas in salivary gland cells, it

is 1.5-fold higher, and in NCL cells it is at least 2.4-fold higher. $Ca^{2+}$ tunneling has also been shown to regulate the frequency of $Ca^{2+}$ signals where it favors tonic over oscillatory $Ca^{2+}$ transients (Courjaret et al., 2017).

Collectively, our data show that store depletion remodels the $Ca^{2+}$ signaling machinery in the cell cortex into subdomains both laterally in the plane of the ER and PM, and axially within the cortical ER to support $Ca^{2+}$ tunneling in delivering $Ca^{2+}$ entering the cell through SOCE to distal effectors. This tunneling mechanism is important functionally in activating $Cl^-$ secretion and sweat production.

## Materials and methods
### Cell culture and solutions
Hela cells (CCL-2; ATCC) were cultured in DMEM media containing 10% fetal bovine serum (FBS) supplemented with penicillin (100 U $ml^{-1}$) and streptomycin (100 μg $ml^{-1}$). The cells were plated 24 h before transfection on poly-lysine coated glass-bottom dishes (MatTek). NCL-SG3 cells were a gift from Stefan Feske (New York University, New York, NY, USA) (Concepcion et al., 2016), and were cultured in Williams E Media supplemented with 5% FBS, with penicillin (100 U $ml^{-1}$), streptomycin (100 μg $ml^{-1}$), glutamine (4 mM), insulin (10 mg $l^{-1}$), transferrin (5.5 mg $l^{-1}$), selenium (6.7 μg $l^{-1}$), hydrocortisone (10 mg $l^{-1}$), and epidermal growth factor (20 μg $l^{-1}$). For live cell experiments, cells were perfused using a peristaltic pump (Gilson Minipuls) at the speed of 1 ml $min^{-1}$. The standard saline contained (in mM) 145 NaCl, 5 KCl, 2 $CaCl_2$, 1 $MgCl_2$, 10 glucose, and 10 HEPES, pH 7.2, for $Ca^{2+}$-free experiments, and the $Ca^{2+}$ was exchanged equimolarly with $Mg^{2+}$.

### Plasmids and transfection
Transfection was performed using Lipofectamine 2000 (Thermo Fisher Scientific) according to the manufacturer's instructions. mCherry-STIM1 and GFP-Orai1 were a gift from Rich Lewis (Stanford University, Stanford, CA, USA) (Park et al., 2009), EGFP-rIP3R1 from Colin Taylor (Cambridge University, Cambridge, UK) (Pantazaka and Taylor, 2011), GFP-Mapper from Jen Liou (UT Southwestern, Dallas, TX, USA) (Chang et al., 2013), and Ano1–GFP from Karl Kunzelmann (Regensburg, Germany) (Cabrita et al., 2017). EGFP-hPMCA4b (Chicka and Strehler, 2003) and the $Cl^-$ reporter mbYFPQS (Watts et al., 2012) were obtained from Addgene (#47589 and #80742) and PLN-GFP from Origene (#RG202712). CaTAr1 and 2 were custom synthesized by Genewiz and sequence verified. CaTArΔpolyK, lacking the N-terminal poly-lysine domain from CaTAr1, was custom-synthesized by GeneUniversal and sequence-verified.

### Intracellular $Ca^{2+}$ imaging
To image cytoplasmic $Ca^{2+}$, cells were loaded for 30 min at room temperature with either 2 μM Calbryte 590 AM (#20700; AAT Bioquest) or Fluo4-AM (F14201; Thermo Fisher Scientific), 2 mM stocks were made in 20% pluronic acid/DMSO. Imaging was performed on a Zeiss LSM880 confocal system fitted with a 40x/1.3 oil immersion objective using an open pinhole at a frame rate of 0.1 Hz. The following parameters were used: for Calbryte, $\lambda_{ex}$ = 561 nm and $\lambda_{em}$ = 566–679 nm; and for Fluo4, $\lambda_{ex}$ = 488 nm

and $\lambda_{em}$ = 493–574 nm. The expression level of either PLN-Map-GFP or SLN-Map-Ch was recorded using z-stacks and a pinhole set to 1AU cells at the beginning of the experiment using the following parameters: $\lambda_{ex}$ = 488 nm and $\lambda_{em}$ = 493–574 (GFP) and $\lambda_{ex}$ = 561 nm and $\lambda_{em}$ = 578–696 nm(mCherry).

## IP$_3$ uncaging

The cells were loaded for 30 min at room temperature with a mixture of 2 µM Calbryte 590 AM and caged IP$_3$ (cag-iso-2-145-10; Sichem). The stores were depleted using the CPA application and the SERCA function was restored by washing out the CPA in a Ca$^{2+}$-free media for 20 min. Extracellular Ca$^{2+}$ was then reintroduced in the extracellular media using the perfusion system and the cytosolic Ca$^{2+}$ levels monitored. We let SOCE start to develop (20 s) and uncaged IP$_3$ using the 405 nm laser line of the confocal using a single scan event.

## Airyscan imaging

High-resolution images were acquired using the Airyscan detector of a Zeiss LSM880 confocal microscope using the super-resolution mode (SR) and default image processing parameters. The 488 and 561 laser lines and a 488/561 MBS were used, and the emitted light was recorded using either a single filter BP495-550/LP570 and a sequential line recording mode or a dual filter protocol (BP495-550/LP570 and BP420-480+495-550) and alternating z-stacks between both wavelengths. Z-stacks were recorded at the recommended intervals (typically 0.18 µm).

## TIRF imaging

TIRF images were acquired on an AxioObserver Z1 microscope (Zeiss) using a 63x/1.46 lens at a maximum angle and using the following parameters: for Alexa 488 and GFP: λex = 488 nm and λem = 510/555; for Alexa 555 and mCherry λex = 561 nm and λem = 581/679.

## Acinar cell isolation and imaging

SMG acinar cells were enzymatically isolated from 2- to 4-mo-old, C57BL/6J mice of both sexes. To isolate acinar cells, glands were extracted, connective tissue was removed, and glands were minced. Cells were placed in oxygenated dissociation media at 37°C for ~30 min with shaking. Dissociation media consisted of Hank's Balanced Salt Solution containing CaCl$_2$ and MgCl$_2$ (HBSS), bovine serum albumin (0.5%), and collagenase Type II (0.2 mg/ml; Worthington). Cells were washed twice in HBSS with 0.5% BSA and resuspended in a HBSS solution containing 0.5% BSA and 0.02% trypsin inhibitor. Cells were then resuspended in imaging buffer (in mM) 10 HEPES, 1.26 CaCl$_2$, 137 NaCl, 4.7 KCl, 5.5 glucose, 1 Na$_2$HPO$_4$, 0.56 MgCl$_2$, at pH 7.4; with 5 µM FURA 2-AM (F1221; Thermo Fisher Scientific) and seeded onto a Cell-Tak coated coverslip to allow attachment of cells. Cells were then perfused with an imaging buffer and stimulated with an agonist. Ca$^{2+}$ imaging was performed using an inverted epifluorescence Nikon microscope with a 40 X oil immersion objective (NA = 1.3). Cells were alternately excited at 340 and 380 nm, and emission was monitored at 505 nm. Images were captured with a digital camera driven by TILL Photonics

software. Image acquisition was performed using TILLVISION software.

## Patch clamp electrophysiology

For measurements of Cl$^-$ currents, acinar cells were allowed to adhere to Cell-Tak coated glass coverslips for 15 min before experimentation. Coverslips were transferred to a chamber containing extracellular bath solution (in mM) 155 tetraethylammonium chloride to block K$^+$ channels, 2 CaCl$_2$, 1 MgCl$_2$, and 10 HEPES; pH 7.2. Ca$^{2+}$-free bath solution substituted 1 mM EGTA for CaCl$_2$. Cl$^-$ currents in individual cells were measured in the whole-cell patch clamp configuration using pClamp 9 and an Axopatch 200B amplifier (Molecular Devices). Recordings were sampled at 2 kHz and filtered at 1 kHz. Pipette resistances were 3–5 MΩ and seal resistances were >1 GΩ. Pipette solutions (pH 7.2) contained (in mM) 60 tetraethylammonium chloride, 90 tetraethylammonium glutamate, 10 HEPES, 1 HEDTA (N-(2-hydroxyethyl) ethylenediamine-N,N',N'-triacetic acid), and 100 nM free Ca$^{2+}$ were used to mimic physiological buffering and basal [Ca$^{2+}$]$_i$ conditions. Free [Ca$^{2+}$] was estimated using Maxchelator freeware. Agonists were directly perfused onto individual cells using a multibarrel perfusion pipette.

## Sweat test

Sweating was measured using the starch/iodine technique (Yu et al., 2022). C57Bl6 male mice 3- to 5-mo-old were used. The animals were injected with a control adenovirus (Ad-GFP, #1060; Vector Biolabs) or an adenovirus expressing CatAr1 (Ad-CMV; Vector Biolabs). The viruses in sterile PBS were injected at 50 µl at a titer of 10$^8$ PFU/ml in the right hind foot pad of anesthetized mice. The virus was allowed to express for 4 days prior to the sweat test. Images were acquired using an ERc5s Axiocam mounted on a STEMI 305 stereo microscope (Zeiss), quipped with 0.5x Front Optics 3 (435263-9050-000). Paw images were taken every 3 min and analyzed using ImageJ. The area covered by the black dots measured after 15 min was used to report sweating levels. Fluorescence images of the foot pads were taken using a fluorescence stereo microscope (Zeiss Lumar V12) equipped with a color camera (Zeiss Axiocam MR5) and a 0.8x Neolumar S lens (435206-9901-000).

## Immunocytochemistry

The cells were plated on glass bottom dishes and fixed using PFA (4%, 10 min), permeabilization was achieved using Triton x100 (10 min, 0.3%), and saturation for 1 h using a mix of 10% goat serum and 1% bovine serum albumin. Primary antibody incubation was performed at 4°C overnight. The following primary antibodies were used at a 1:500 dilution: SERCA2 (NB300-581; Novus Biologicals), STIM1 (5668S; Cell Signaling and MA1-19451; Thermo Fisher Scientific), KRAP (14157-1-AP; Thermo Fisher Scientific), NFAT1 (5861S; Cell Signaling), and ANO1 (14476; Cell Signaling). For IP$_3$R1 detection, a custom-made monoclonal antibody targeting the following peptide: RIGLLGHPPHMNVNPQQPA (ProSci) was used (Baker et al., 2021). For NFAT translocation quantification the cells were treated with thapsigargin (1 µM, 10 min) prior to fixation. Secondary antibodies were goat anti-mouse and anti-rabbit (Thermo Fisher Scientific) coupled to either Ax488 (A-11001) or

Ax555 (A-11034) and used at a 1:2,000 dilution at room temperature for 2 h.

## Data analysis and statistics

The imaging data was quantified using FIJI/ImageJ 1.51n (Schindelin et al., 2012; Schneider et al., 2012) and ZenBlue 2.3 (Zeiss). To rank the effect of the MAPPER, CaTAr1 and 2 on tunneling in HeLa cells, we used the max/min intensity of the GFP or mCherry signal measured at the beginning of the experiment. The bracket from 0% to 10% was considered as control cells and values over 20% expressing cells. 3D reconstructions were performed using Imaris 9.5 (Bitplane). NND analysis was performed using the DiAna plugin (Gilles et al., 2017) and co-localization using EzColocalization (Stauffer et al., 2018). The patch-clamp data was analyzed with Clampfit 10.0 (Molecular Devices). Statistics and data analysis were performed using Graphpad Prism 10.0.1 (GraphPad). Values are given as mean ± S.E.M and statistics were performed using either paired or unpaired Student's $t$ test or ANOVA followed by Tukey's test for multiple comparisons. P values are ranked as follows: *$P < 0.05$, **$P < 0.01$, ***$P < 0.001$, and ****$P < 0.0001$.

## Online supplemental material

Fig. S1 shows store depletion in primary salivary gland cells and STIM1 Orai1 localization in HeLa cells. Fig. S2 shows localization of SERCA2b and mCh-STIM1. Fig. S3 shows the localization of endogenous IP$_3$R1, STIM1, and KRAP. Fig. S4 shows the localization of IP$_3$R1, STIM1, and KRAP. Fig. S5 shows the localization of MAPPER-S relative to STIM1. Fig. S6 shows the localization of MAPPER–GFP, E-Syt2-mCh, and TMEM24-mCh. Fig. S7 shows CaTAr2 inhibits Ca$^{2+}$ tunneling. Fig. S8 shows PLN and CaTAr∆polyK inhibit SERCA. Fig. S9 shows CaTAr1 effects on tunneling and localization of ANO1–GFP and mCh-STIM1. Fig. S10 shows CaTAr2 and ANO1–GFP in NCL cells. Video 1 shows the formation of STIM1 clusters adjacent to MAPPER. Video 2 shows the formation of STIM1 clusters adjacent to CaTAr1.

## Data availability

All data are included in the manuscript or supplemental figures.

## Acknowledgments

We are grateful to the Imaging and Vivarium Cores at Weill Cornell Medicine Qatar (WCMQ) for their support.

This work as well as the Cores are supported by the Biomedical Research Program at WCMQ (BMRP) to K. Machaca, a program funded by the Qatar Foundation, with additional support from NIH R01DE019245 to D.I. Yule. We are grateful to colleagues who contributed clones and reagents as listed in the Materials and methods section. We are also thankful to Jen Liou for helpful discussions during the early stages of this work regarding tunneling inhibition, and to Sandip Patel for suggesting the acronym CaTAr for the tunneling inhibitor during a presentation at a European Ca$^{2+}$ Society (ECS) meeting.

Author contributions: R.J. Courjaret: Conceptualization, Data curation, Formal analysis, Investigation, Methodology, Project administration, Validation, Visualization, Writing - original draft, Writing - review & editing, L.E. Wagner: Investigation, R.R. Ammouri: Investigation, D.I. Yule: Formal analysis, Funding acquisition, Project administration, Supervision, K. Machaca: Conceptualization, Data curation, Formal analysis, Funding acquisition, Project administration, Supervision, Validation, Visualization, Writing - original draft, Writing - review & editing.

Disclosures: The authors declare no competing interests exist.

Submitted: 18 February 2024

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

# Supplemental material

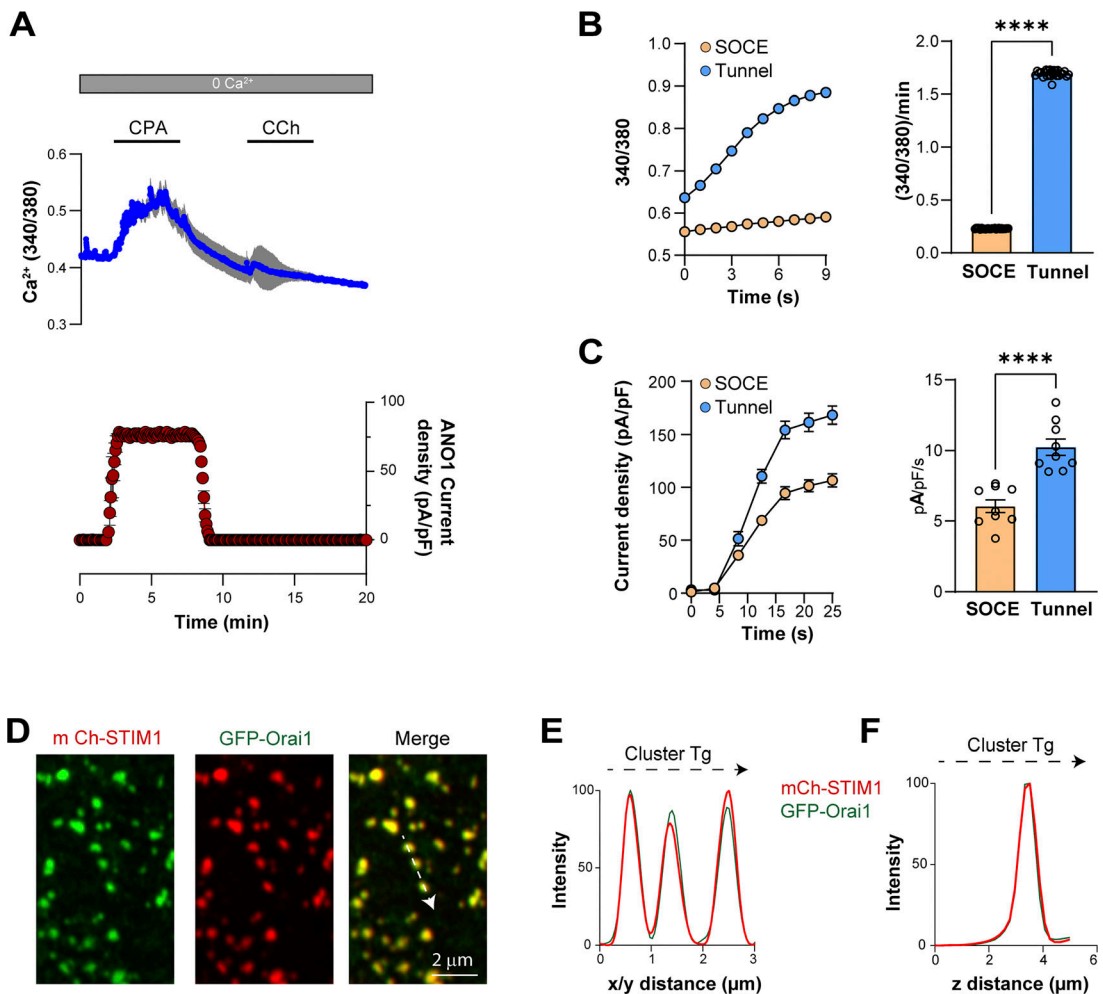

Figure S1. **Store depletion in primary salivary gland cells and STIM1 Orai1 localization in HeLa cells. (A)** The transient application of CPA (30 μM) depletes ER Ca²⁺ stores as indicated by the rise in cytosolic Ca²⁺ (blue) which activates the Cl⁻ current (red). Following the washout of CPA, the application of carbachol (CCh, 10 μM) fails to elicit a response, indicating the effective depletion of the ER stores ($n$ = 4–8). **(B and C)** Kinetics of intracellular Ca²⁺ elevation and Cl⁻ current development during tunneling compared to SOCE ($n$ = 26 (B) and 9 (C); unpaired $t$ test). **(D)** Enlarged confocal images show the colocalization of STIM1 and Orai1 at the PM focal plane. **(E and F)** Plots from a virtual line scan drawn across clusters in the PM plane (E, arrow in D) and in the z axis (F).

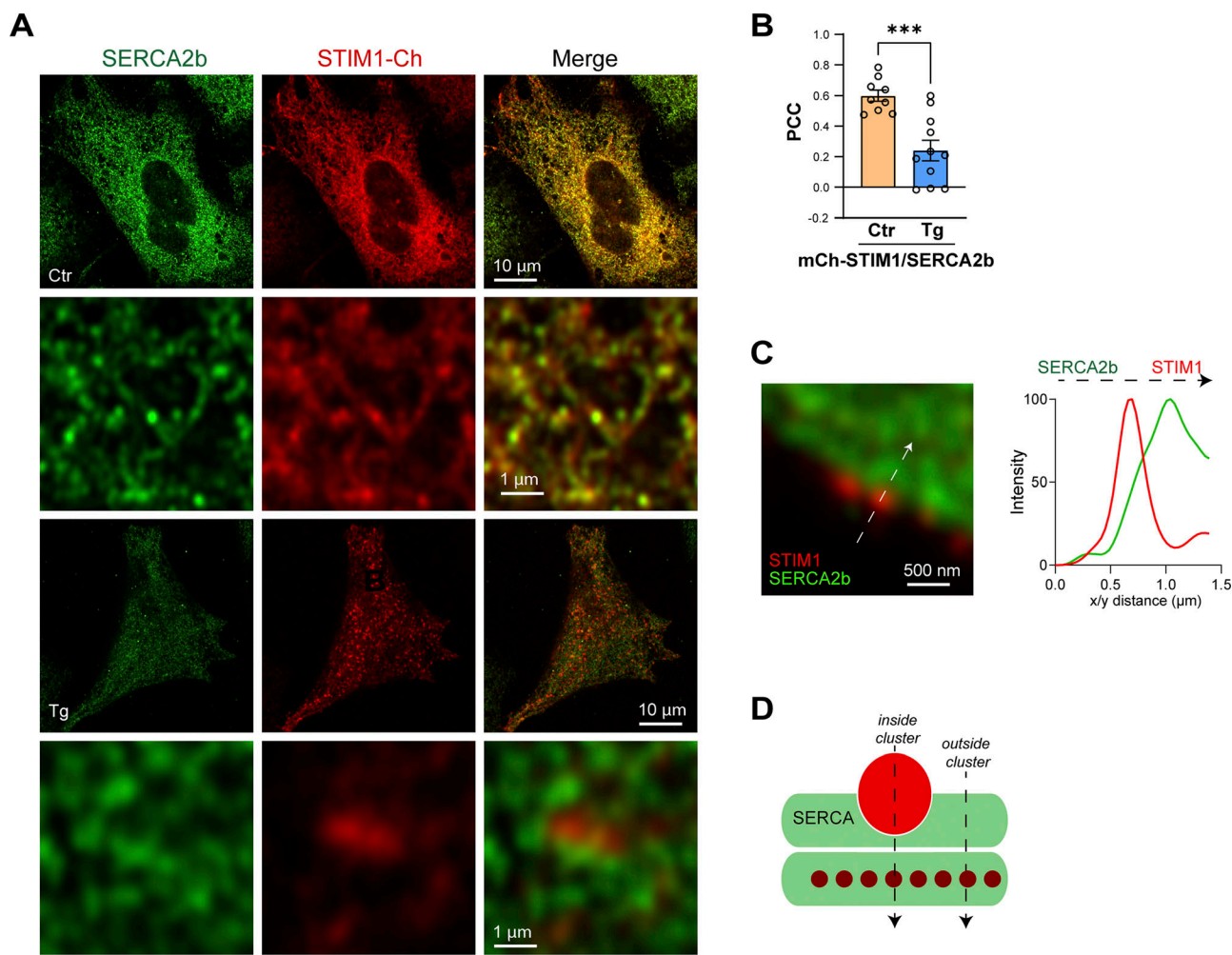

Figure S2. **Localization of SERCA2b and mCh-STIM1. (A)** HeLa cells transfected with mCh-STIM1 and stained using a SERCA2b antibody. AiryScan images were taken before (Ctr) and after store depletion (Tg). Control images are acquired at a focal plane located in the middle of the cell and store-depleted images at the PM plane, where the STIM1 clusters localize. At rest, STIM1 and SERCA2b colocalize in the ER cisternae, while after store depletion they separate from each other at the PM focal place. **(B)** Pearson's correlation coefficient (PCC) obtained before (Ctr) and after store depletion (Tg) between endogenous SERCA2b and overexpressed mCh-STIM1. Colocalization at rest was measured in the middle of the cell and at the PM plane after store depletion ($n$ = 9–11; unpaired $t$ test). **(C)** Image and line scan of STIM1 clusters on the side of the cell depicting its isolation from SERCA2b. **(D)** Cartoon illustrating the localization of STIM1 (red) and SERCA (green) after store depletion. The positions of the line scans in Fig. 2 H are also shown.

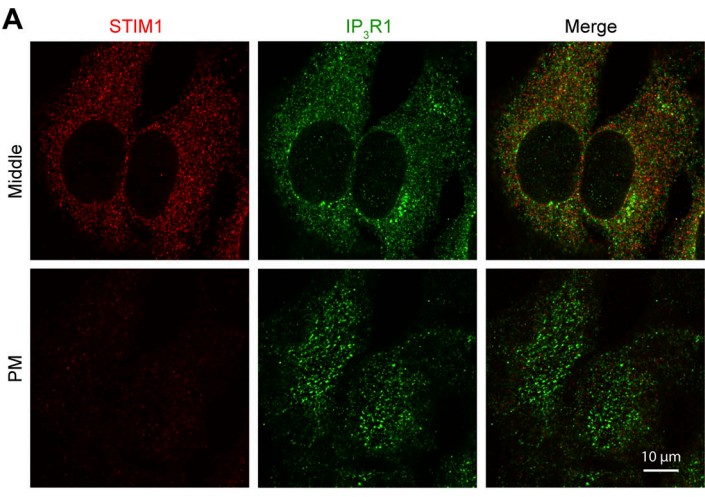

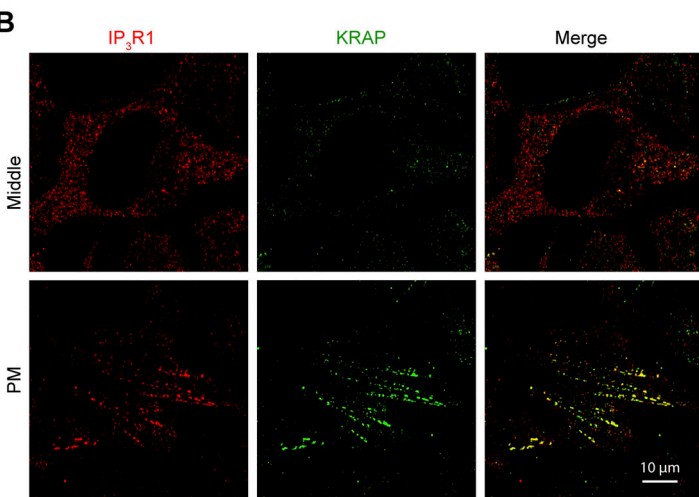

Figure S3.   **Localization of endogenous IP₃R1, STIM1 and KRAP. (A)** Relative localization of STIM1 and IP₃R1 detected by immunofluorescence in HeLa cells at rest. While both proteins share the same intracellular compartment, there is no overlap of the signals at the PM where the IP₃R1 fluorescence reveals the "licensed" receptors. **(B)** Colocalization at the PM focal plane of IP₃R1 and KRAP in HeLa cells at rest.

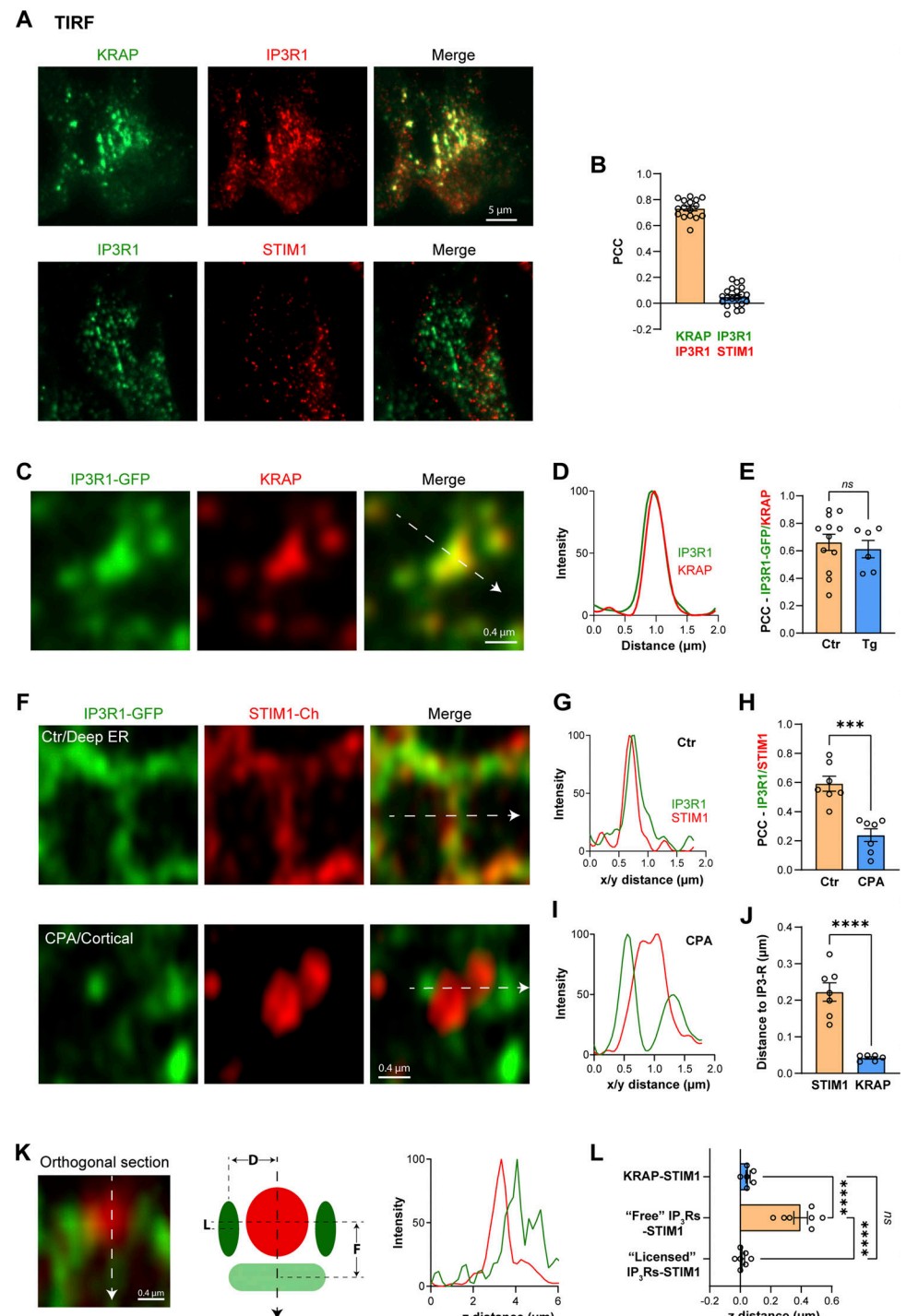

Figure S4.  **Localization of IP₃R1, STIM1, and KRAP. (A)** TIRF images of STIM1, KRAP, and IP₃R1 were detected by immunofluorescence in HeLa cells after store depletion. **(B)** Bar chart summarizing colocalization (PCC) of IP₃R1 with KRAP but not with STIM1 (*n* = 16–21; unpaired *t* test). **(C)** Airyscan images or "licensed" IP₃R1–GFP and KRAP detected by immunofluorescence at the PM plane (*n* = 6–12; unpaired *t* test). **(D)** Relative intensities were measured along the line indicated by the white arrow in C. **(E)** Colocalization (PCC) between KRAP and IP₃R1–GFP before (Ctr) and after store depletion (Tg). **(F)** Airyscan images of IP₃R1–GFP and mCh-STIM1 inside the cell (deep ER, in control conditions) and at the PM after store depletion (CPA/Cortical). **(G)** Relative intensities measured along the line indicated by the white arrow in F in cells at rest. **(H)** Colocalization (PCC) of IP₃R1–GFP and mCh-STIM1 before and after store depletion (CPA) (*n* = 7; paired *t* test). **(I)** Relative intensities along the line indicated by the white arrow in F in store depleted cells. **(J)** Lateral distance between the mCh-STIM clusters and either IP₃R1–GFP or endogenous KRAP after store depletion (*n* = 6–7; unpaired *t* test). **(K)** Example of an orthogonal section through a mCh-STIM cluster highlighting the localization of "licensed" IP₃R1–GFP outside the cluster and "Free" IP₃R deeper within the ER away from the STIM1 cluster as illustrated by the intensity plot. The cartoon indicates the distribution of STIM1 (red) after store depletion relative to "Licensed" (L) and "Free" (F) IP₃R1–GFP. The lateral distance between STIM1 clusters and licensed IP₃R1 as measured in panel J is indicated as D. **(L)** Axial distance between STIM1 clusters after store depletion and KRAP, "Free" IP₃R1–GFP, or "Licensed" IP₃R1–GFP, as indicated. "Free" IP₃R1 are receptors that localized deeper in the cell and do not colocalize with KRAP.

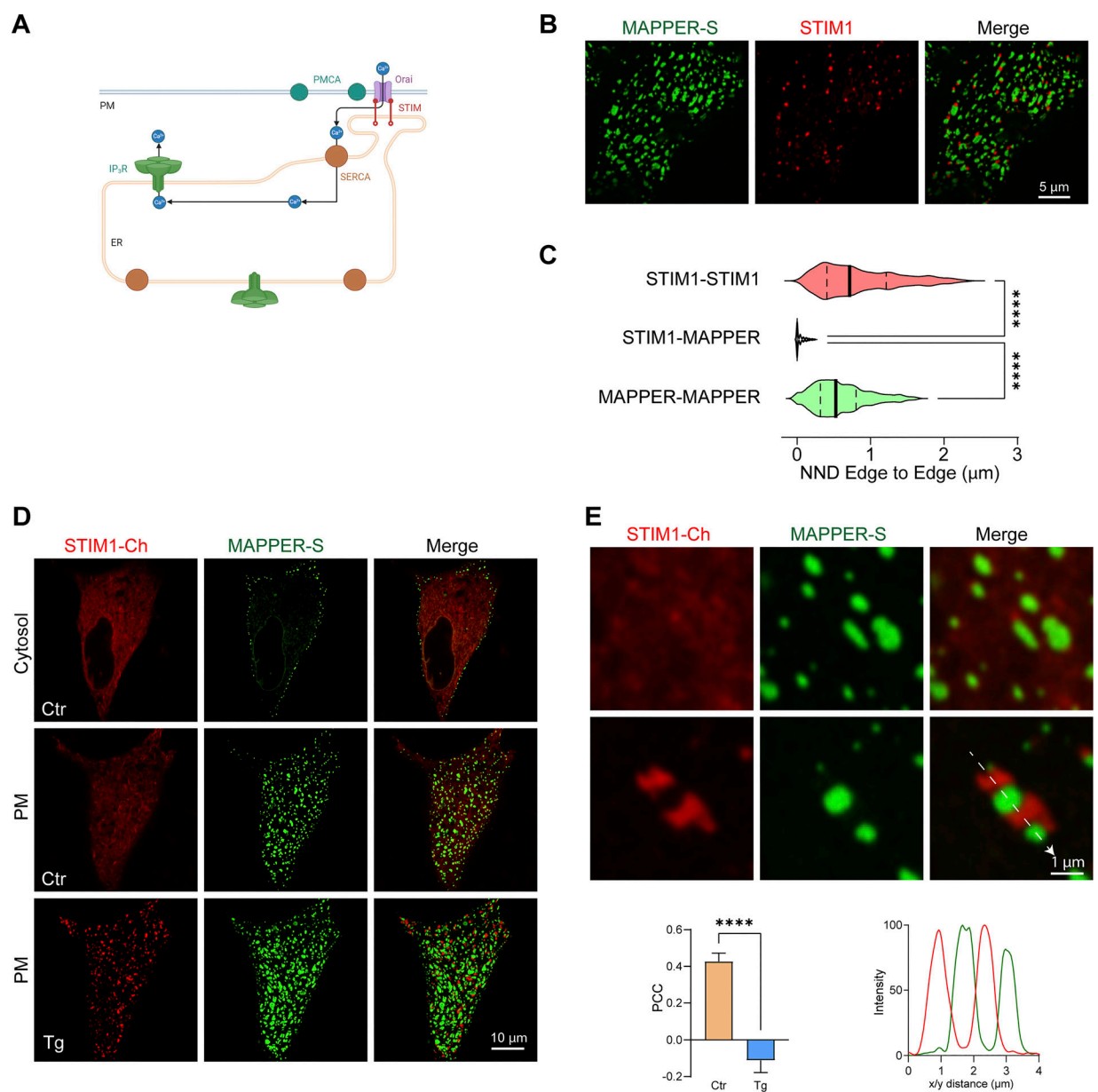

Figure S5.   **Localization of MAPPER-S relative to STIM1. (A)** Cartoon illustrating the spatial organization of the tunneling machinery. **(B)** Airyscan images of MAPPER–GFP (MAPPER) and mCh-STIM1 following store depletion with CPA in HeLa cells show that they do not colocalize. **(C)** Violin plots showing the nearest neighbor distance (NND) between the edges of the STIM1 and MAPPER clusters ($n$ = 626–1,585, outliers removed using the ROUT routine, one-way ANOVA). **(D)** Relative localization of mCh-STIM1 and short MAPPER (MAPPER-S) in control conditions (Ctr) and after store depletion (Tg) at the whole cell level. **(E)** High magnification images of mCh-STIM1 and MAPPER-S clusters with the corresponding PCC ($n$ = 11, paired $t$ test) and line scan profile measured along the white arrow.

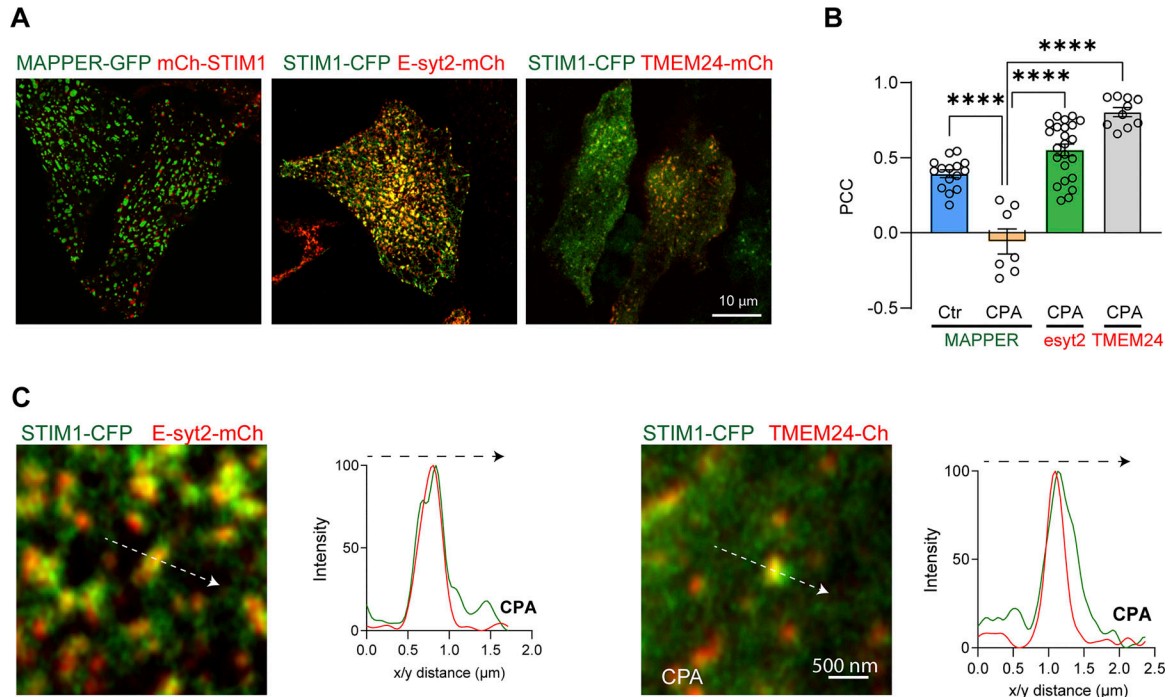

Figure S6. **Localization of MAPPER–GFP, E-Syt2-mCh, and TMEM24-mCh. (A)** Localization of MAPPER–GFP, E-Syt2-mCh, and TMEM24-mCh at the whole cell level. **(B)** Bar chart summarizing the PCC of the three tethers relative to STIM1 ($n$ = 7–23; one-way ANOVA). **(C)** Confocal images and line scans illustrate the colocalization of E-Syt2-mCh and TMEM24-mCh with STIM1-CFP clusters. The intensity plots are obtained from the lines depicted by the white arrows.

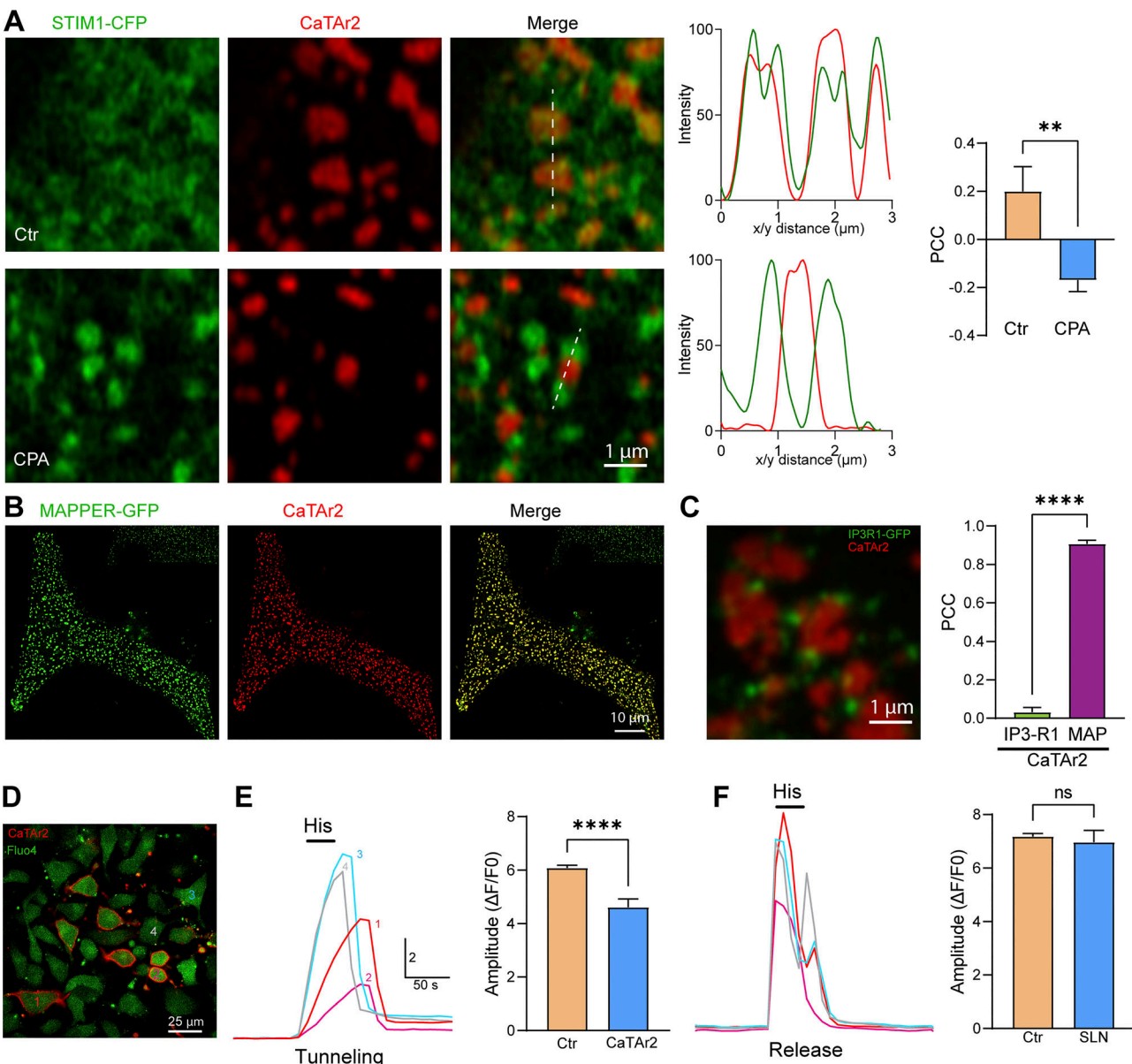

Figure S7. **CaTAr2 inhibits Ca²⁺ tunneling. (A)** Localization of CaTAr2 relative to STIM1-CFP before (Ctr) and after store depletion (CPA). Colocalization analysis using either intensity plots along the white line or PCC measurements confirms that CaTAr2 localizes in a similar fashion to CaTAr1 and MAPPER relative to STIM1 clusters ($n$ = 5–10; unpaired $t$ test). **(B)** Colocalization of MAPPER–GFP and CaTAr2 at the whole cell level. **(C)** Localization and PCC between CaTAr2 and IP₃R1–GFP. The colocalization of CatAr2 with MAPPER was used as a reference value for the PCC analysis ($n$ = 6–14; unpaired $t$ test). **(D)** Confocal images of HeLa cells expressing CaTAr2 and loaded with Fluo4-AM. The numbered cells refer to the traces in E and F. **(E)** Ca²⁺ tunneling traces obtained after store depletion with CPA indicate that cells expressing CaTAr2 have a lower tunneling capacity. The bar chart on the right summarizes the inhibition of Ca²⁺ tunneling by CaTAr2. **(F)** Ca²⁺ release traces were obtained on the same cells as in E by applying histamine (His, 100 µM) after store refilling. The bar chart on the right summarizes the effect of CaTAr2 on Ca²⁺ release ($n$ = 42–342; unpaired $t$ test).

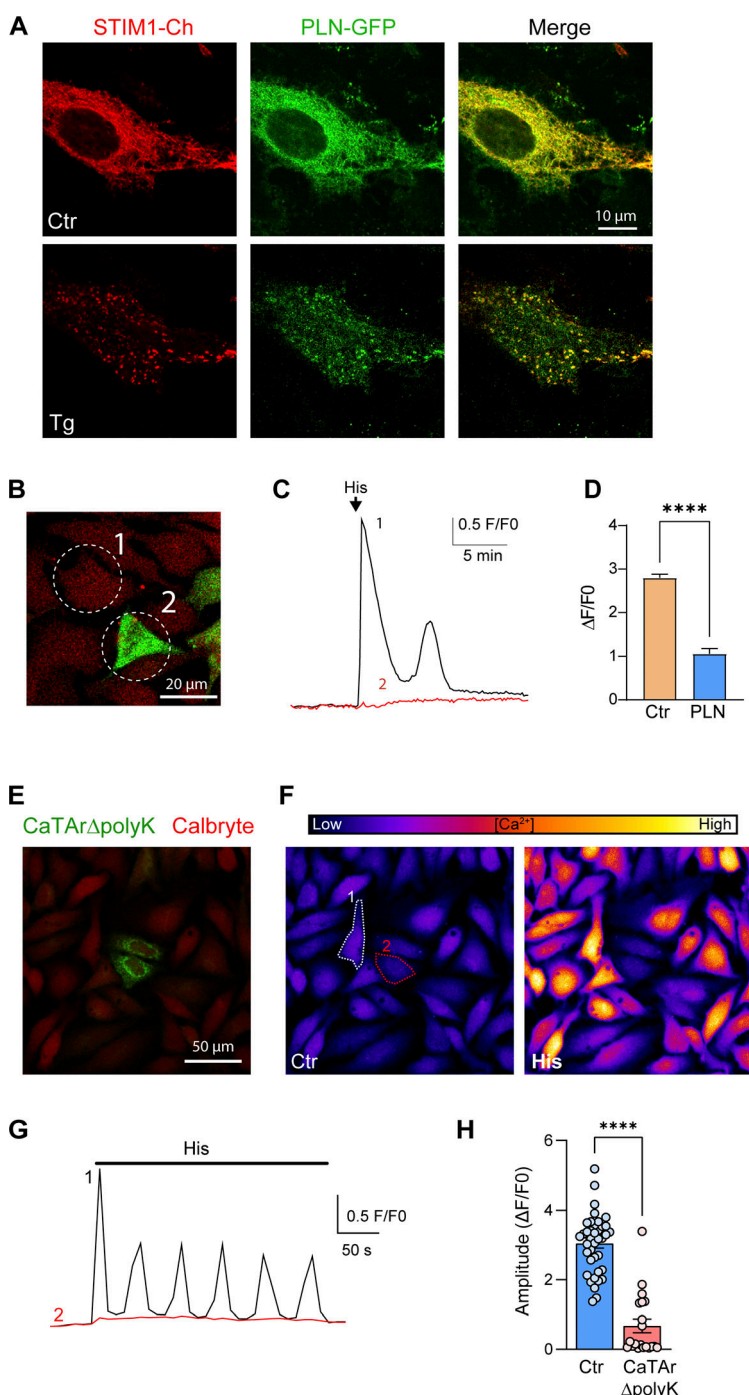

Figure S8. **Phospholamban and CaTArΔpolyK inhibit SERCA. (A)** Localization of phospholamban-GFP (PLN-GFP) and mCh-STIM1 in HeLa cells before (Ctr) and after store depletion (Tg). **(B)** PLN-GFP transfection followed by loading cells with Calbryte 590. The numbered cells match the traces in C. **(C)** Ca²⁺ release induced by histamine (His, 100 μM) leads to Ca²⁺ release in cell#1, which does not express PLN-GFP; but not in cell#2, which expresses high levels of PLN-GFP. **(D)** Bar chart summarizing the inhibition of Ca²⁺ release in PLN-expressing cells. The recordings were performed in the presence of 1 mM lanthanum in the extracellular media to inhibit Ca²⁺ influx and recycling at the PM ($n$ = 38–176; unpaired $t$ test). **(E)** ER localization of CaTArΔpolyK (green), a CaTAr1 construct lacking the terminal poly-lysine domain that localizes it to ERPMCS. **(F)** Example of Calbryte intensity images before (Ctr) and after histamine addition (His). **(G)** Ca²⁺ release traces induced by histamine (His, 100 μM) in the two cells labeled in panel F. **(H)** Bar chart summarizing the inhibition of Ca²⁺ release in CaTArΔpolyK-expressing cells ($n$ = 20–39; unpaired $t$ test; P < 0.0001).

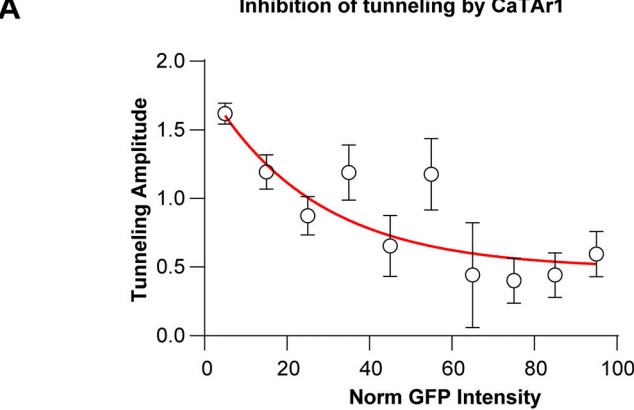

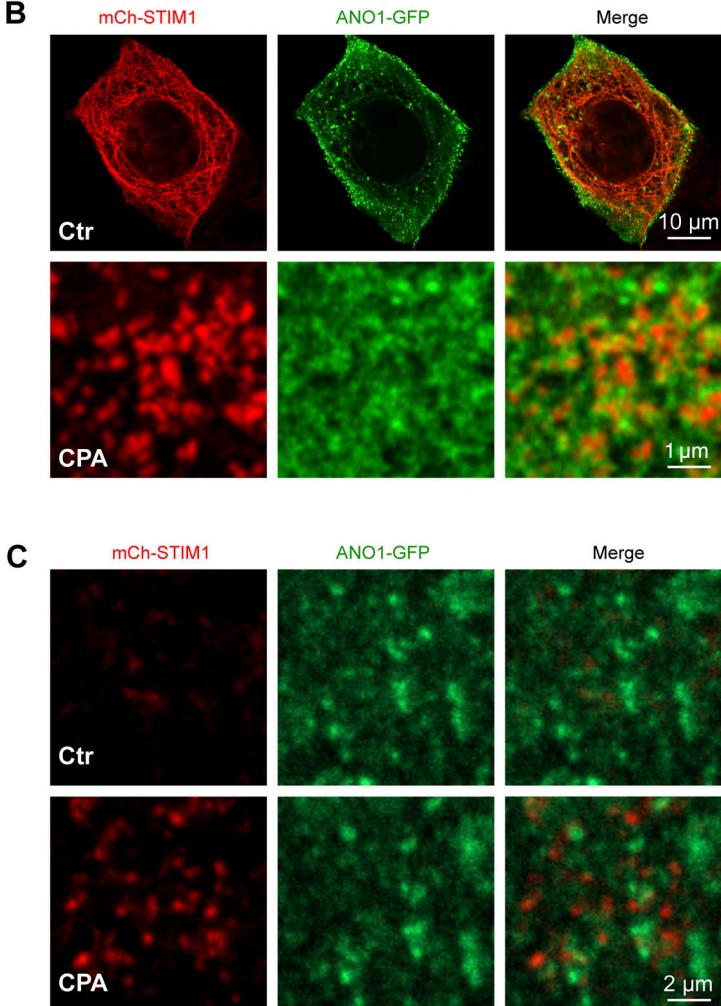

Figure S9. **CaTAr1 effects on tunneling and localization of ANO1–GFP and mCh-STIM1. (A)** Dose-dependent inhibition of tunneling by CaTAr1. The amplitude of the tunneling signal is plotted as a function of the GFP signal normalized from minimum to maximum signal in each dish. The data is sorted in bins of 10%. **(B)** The relative localization of ANO1–GFP and mCh-STIM1 in NCL cells after store depletion is similar to the endogenous ANO1 channel and suggests the separation of ANO1 from the STIM1 cluster. **(C)** TIRF imaging of ANO1–GFP and mCh-STIM1 during store depletion with CPA confirmed the separation of the two proteins at the PM. Ctr and CPA images are from the same region of interest.

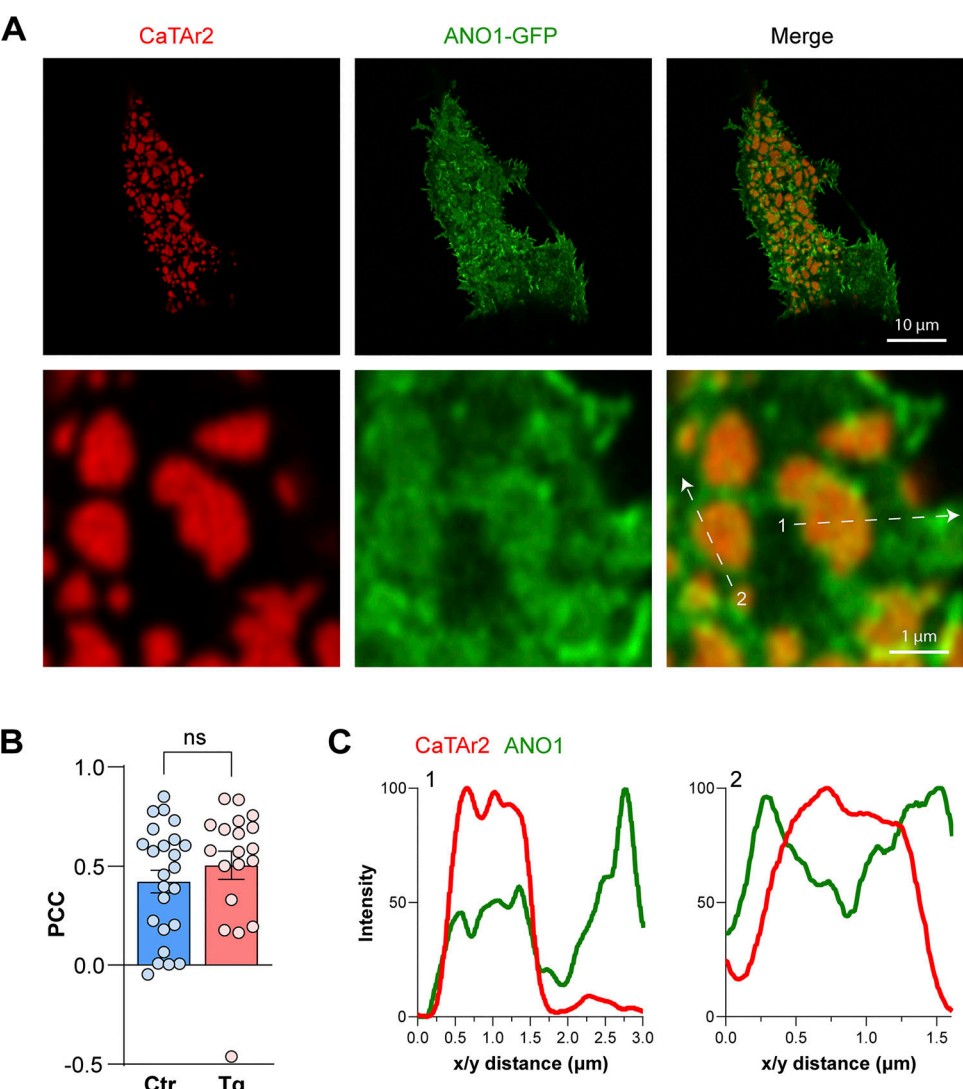

Figure S10. **CaTAr2 and ANO1–GFP in NCL cells. (A)** Airy scan images of the Co-expression of CaTAr2 and ANO1–GFP in NCL cells indicate the partial overlap between the two signals. **(B)** The colocalization is not influenced by store depletion. **(C)** Relative intensities along a virtual line scan (white arrows in A) reveal the colocalization of ANO1 and CaTAr2 but also enrichment in ANO1 at the edge of the CaTAr2-rich ER patch.

Video 1. **Formation of STIM1 clusters adjacent to MAPPER.** A HEK293 cell overexpressing MAPPER–GFP (green) and mCh-STIM1 (red) is imaged over time using TIRF microscopy. Store depletion is induced using thapsigargin leading to the formation of STIM1 clusters next to MAPPER–GFP contact sites. The frame rate is 1.2 Hz for a total duration of 10 min.

Video 2. **Formation of STIM1 clusters adjacent to CaTAr1.** A HeLa cell overexpressing CaTAr1 (green) and mCh-STIM1 (red) is imaged over time using TIRF microscopy. Store depletion is induced using thapsigargin leading to the formation of STIM1 clusters next to CaTAr1 expression sites. The frame rate is 0.1 Hz for a total duration of 36 min.

