## [Peer Review File · The Journal of Cell Biology]

Ca²⁺ tunneling architecture and function are important for secretion

Raphael Courjaret, Larry Wagner II, Rahaf Ammouri, David Yule, and Khaled Machaca

Corresponding Author(s): Khaled Machaca, Weill Cornell Medical College in Qatar

Review Timeline:

Submission Date:	2024-02-18
Editorial Decision:	2024-04-02
Revision Received:	2024-06-29
Editorial Decision:	2024-08-22
Revision Received:	2024-09-29
Editorial Decision:	2024-10-01
Revision Received:	2024-10-08

Monitoring Editor: William Prinz

Scientific Editor: Dan Simon

Transaction Report:

DOI: <https://doi.org/10.1083/jcb.202402107>

April 2, 2024

Re: JCB manuscript #202402107

Prof. Khaled Machaca
Weill Cornell Medical College in Qatar
Luqta Street
Education City
Doha 24144
Qatar

Dear Prof. Machaca,

Thank you for submitting your manuscript entitled "Architecture of Ca²⁺ tunneling, a basic Ca²⁺ signaling modality important for secretion." Your manuscript has been assessed by expert reviewers, whose comments are appended below. Although the reviewers express potential interest in this work, significant concerns unfortunately preclude publication of the current version of the manuscript in JCB.

You will see that Reviewer #3 is very enthusiastic and only asks for text edits to improve the accessibility of your work for a broad audience. Reviewers #1&2 are also supportive but request new experiments that we agree would be important to add. These include data showing that activation of ionotropic receptors and different sensitivities of the IP3 receptor do not cause Ca²⁺ release from the ER, additional controls for the CaTAr inhibitors, assays of Ca²⁺ tunnelling under PLC activation, improved statistical analysis, and clarification of several results.

Please let us know if you are able to address the major issues outlined above and wish to submit a revised manuscript to JCB. Note that a substantial amount of additional experimental data likely would be needed to satisfactorily address the concerns of the reviewers. The typical timeframe for revisions is three to four months. While most universities and institutes have reopened labs and allowed researchers to begin working at nearly pre-pandemic levels, we at JCB realize that the lingering effects of the COVID-19 pandemic may still be impacting some aspects of your work, including the acquisition of equipment and reagents. Therefore, if you anticipate any difficulties in meeting this aforementioned revision time limit, please contact us and we can work with you to find an appropriate time frame for resubmission. Please note that papers are generally considered through only one revision cycle, so any revised manuscript will likely be either accepted or rejected.

If you choose to revise and resubmit your manuscript, please also attend to the following editorial points. Please direct any editorial questions to the journal office.

GENERAL GUIDELINES:

Text limits: Character count is < 40,000, not including spaces. Count includes title page, abstract, introduction, results, discussion, and acknowledgments. Count does not include materials and methods, figure legends, references, tables, or supplemental legends.

Figures: Your manuscript may have up to 10 main text figures. To avoid delays in production, figures must be prepared according to the policies outlined in our Instructions to Authors, under Data Presentation, <https://jcb.rupress.org/site/misc/ifora.xhtml>. All figures in accepted manuscripts will be screened prior to publication.

Supplemental information: There are strict limits on the allowable amount of supplemental data. Your manuscript may have up to 5 supplemental figures. Up to 10 supplemental videos or flash animations are allowed. A summary of all supplemental material should appear at the end of the Materials and methods section.

Please note that JCB requires authors to submit Source Data used to generate figures containing gels and Western blots with all revised manuscripts. This Source Data consists of fully uncropped and unprocessed images for each gel/blot displayed in the main and supplemental figures. If your revised paper will include cropped gel and/or blot images, please be sure to provide one Source Data file for each figure that contains gels and/or blots along with your revised manuscript files. File names for Source Data figures should be alphanumeric without any spaces or special characters (i.e., SourceDataF#, where F# refers to the associated main figure number or SourceDataFS# for those associated with Supplementary figures). The lanes of the gels/blots should be labeled as they are in the associated figure, the place where cropping was applied should be marked (with a box), and molecular weight/size standards should be labeled wherever possible. Source Data files will be made available to reviewers

during evaluation of revised manuscripts and, if your paper is eventually published in JCB, the files will be directly linked to specific figures in the published article.

If you choose to resubmit, please include a cover letter addressing the reviewers' comments point by point. Please also highlight all changes in the text of the manuscript.

Regardless of how you choose to proceed, we hope that the comments below will prove constructive as your work progresses. We would be happy to discuss them further once you've had a chance to consider the points raised. You can contact the journal office with any questions at cellbio@rockefeller.edu.

Thank you for thinking of JCB as an appropriate place to publish your work.

Sincerely,

William Prinz, PhD
Monitoring Editor
Journal of Cell Biology

Dan Simon, PhD
Scientific Editor
Journal of Cell Biology

Reviewer #1 (Comments to the Authors (Required)):

This paper by Raphael Courjaret et al. investigates the mechanism of Ca²⁺ signal propagation in exocrine secretory cells. In previous work, the Authors described a new Ca²⁺ signaling modality, namely a form of Ca²⁺ signal propagation facilitated by diffusion within the ER lumen, as opposed to propagating simply through the cytoplasm. They have called this mechanism "Ca²⁺ tunneling". In this model, Ca²⁺, entering via the store-operated Ca²⁺ entry (SOCE) pathway at ER-plasma membrane (PM) contact sites is rapidly pumped into the ER lumen by the adjacent SERCA Ca²⁺ pumps, and the Ca²⁺ in the ER lumen is then released by active IP₃ receptors that might be located further away from the Ca²⁺ entry sites. The Authors argue that tunneling is a more efficient way of propagating Ca²⁺ signal away from Ca²⁺ entry sites than simple cytoplasmic diffusion. In this study, they performed high resolution image analysis to estimate the relative localization of the molecular players contributing to tunneling (SERCA and PMCA Ca²⁺ pumps, IP₃ receptors and ANO1 Cl⁻ channels, all relative to STIM1/Orai1). They also attempted to create an experimental approach to specifically interfere with Ca²⁺ tunneling by reducing SERCA pump activity selectively adjacent to the STIM1/Orai1-marked Ca²⁺ entry sites.

Using Airyscan imaging of either endogenous proteins immunostained in fixed cells, or proteins expressed with fluorescent tags, the Authors show that after store depletion, STIM1 puncta do not overlap with the SERCA2b Ca²⁺ pumps, and the latter is found (enriched?) in the adjacent ER regions. Similarly, IP₃Rs, whether the peripheral ones (called "licensed") that form clusters together with the recently identified KRAP protein, or the simply ER-localized diffuse IP₃R pool, show no co-localization with STIM1. IP₃Rs are generally found at some distance from the STIM1-marked Ca²⁺ entry sites. The Authors found no specific enrichment of the PM Ca²⁺ pump, PMCA within the PM after ER Ca²⁺ store depletion. They also performed EM studies for 3D reconstruction of the ER compartment at ER-PM contacts.

In an attempt to inhibit the SERCA pumps adjacent to the STIM1/Orai1 Ca²⁺ entry sites, the Authors used a design based on the ER-PM marker, MAPPER, described previously by the Liou group. For this, they replaced the STIM1 transmembrane (TM) domain with the TM domain of either the PLN or SLN, two proteins that inhibit SERCA via association of their TM domains with the SERCA pump. This design keeps these constructs in preformed ER-PM contact sites in resting cells but because of their limited linker length, they are "squeezed out" from the STIM1/Orai1 clusters during store depletion. Using this tool, which the Authors named CaTar, they show (in HeLa cells) that CaTar transfection interferes with Ca²⁺ tunneling without inhibiting histamine-induced Ca²⁺ release, or store-operated Ca²⁺ entry, the latter judged by NFAT nuclear translocation. The Authors extended their studies further to human sweat cells and showed that activation of distant Cl⁻ channels during stimulation of the PAR2 receptors was severely impaired when CaTar was expressed. Finally, the Authors showed that sweat responses in the paw of mice are inhibited after injection of an adenovirus-vector based CaTar expression plasmid in the paws of the animal. The Authors conclude that "Ca²⁺ tunneling" is critical for the activation of Ca²⁺ effectors that are located at a distance away from SOCE entry sites, such as the ANO1 chloride channels and this mechanism is physiologically important in the regulation of sweat glands both in vivo and in vitro models.

These are interesting studies, extending the Authors previous observations. Analysis of the structural arrangement as well as showing the physiological relevance of Ca²⁺ tunneling advance our knowledge on various modes of Ca²⁺ signal propagation.

While I appreciate the clever design of the CaTAR construct, I feel that there are important controls missing to validate the conclusions. Moreover, the Authors appear to ignore the important difference between ER store depletion by Tg and by PLC activation, as the latter also generates lipid changes in the PM in addition to generating IP3. I also question some of the conclusions based on the immunostaining of SERCA2b.

Specifically:

1. SERCA2b enrichment is not convincing around STIM1 clusters within the ER after Tg treatment. In fact, Fig. S2 shows more SERCA2b clusters in control cells that seem to dissipate after ER store depletion.
2. The Authors use thapsigargin (Tg) to activate SOCE in all confocal experiments to show the relative distribution of the putative molecular players. However, "Ca²⁺ tunneling", by definition, happens when IP3 is also generated, that is during PLC activation. PLC activation, however, also generates another set of messengers, namely diacylglycerol (DAG) and phosphatidic acid (PA). These messengers would activate other effectors, such as PKC or the PIP transfer proteins, Nir2/Nir3, which can have an impact on how Ca²⁺ signals are generated or propagated. I feel that this angle of the story is completely overlooked. It would be informative to use caged IP3 instead of agonist stimulation to assess Ca²⁺ tunneling without the other effects of PLC activation that inevitably adds a further layer of complexity.
3. While the 3D reconstruction of the PM-adjacent ER is impressive, I do not see its value as it relates to the main message of the paper. Perhaps immuno-detection of the SERCA pump in these images relative to STIM1/Orai1 clusters would make this EM analysis a lot more relevant.
4. The design of CaTAR is really clever, but it has some caveats. Namely, this construct has a polybasic domain at the C-terminus that interacts with the negatively charged lipids, PI4P and PI(4,5)P2 in the plasma membrane. This construct, indeed, is pushed to the periphery of the STIM1/Orai1 clusters upon ER store depletion, when Tg was used to empty the Ca²⁺ stores as its interaction with the PM lipids is still maintained. However, when the cells are activated by a PLC agonist, PI(4,5)P2 and PI4P levels may drop, which limits the association of the polybasic domain with the PM and could allow CaTAR to diffuse away from STIM1-adjacent ER regions. The Authors should have shown the localization of this construct (as well as some of the other molecular players) after PLC activation and not only after Tg treatment.
5. The Authors used the expression of the full-length PLN protein to show its inhibition of the SERCA pump when expressed. However, a better control would be to show that the CaTAR construct exerts a global SERCA inhibition when released from the PM contacts by removing its polybasic domain.

Reviewer #2 (Comments to the Authors (Required)):

The study by Courjaret et al. uses high-resolution fluorescence imaging to map the redistribution of calcium transporters along membrane contact sites (MCS) during store-operated calcium entry (SOCE) mediated by interactions between STIM1 on the ER and ORAI1 channels at the PM. They show that SERCA and the artificial tether MAPPER redistribute laterally and axially as STIM1 interacts with ORAI1 at MCS and exploit this co-segregation to generate a local inhibitor of MCS-bound SERCA by replacing the MAPPER TM domain with the TM domain of inhibitory proteins such as phospholamban. The chimera prevent the activation of calcium-activated chloride channels located far from Ca²⁺ entry sites, reduces chloride secretion in primary acinar cells, and reduces sweating upon viral injection in mice. The authors conclude that calcium diffusion inside cortical ER structures expands the spatial range of calcium signals and that this "calcium tunnelling" signaling modality is important for fluid secretion.

Comments:

The concept that calcium diffuses inside the lumen of the endoplasmic reticulum to rapidly reach distant effector target is supported by an array of functional and structural evidence largely gathered by the Machaca group who formulated the "calcium tunnelling" hypothesis. This study provides direct experimental proof that local ER refilling at plasma membrane contact sites drives the tunnelling process. The approach used to achieve a local inhibition of SERCA pumps at sites of contact between the ER and PM is very elegant and the novel molecular tools provide direct functional evidence that the tunnelling process sustains chloride secretion in salivary glands and fluid secretion by sweat glands. The high-quality imaging and innovative functional approaches clearly establish the physiological relevance of calcium tunnelling in fluid secretion. My enthusiasm is somewhat limited by the rather redundant presentation of the first figures and the lack of control experiments for the validation of the genetically encoded SERCA inhibitors. I also have concerns with the statistical analysis as detailed below.

The functional evidence that calcium is "tunnelling" relies on the presence of Ca²⁺ responses when agonists that fail to evoke a response in calcium free medium are applied immediately during calcium readmission (Fig. 1 and S1). The lack of response in Ca²⁺-free is taken as proof that the tunnelling response does not reflect calcium release from the ER. I have two concerns here. 1) did the authors exclude the possibility that the agonist could activate ionotropic receptors? These would generate similar responses entirely dependent on external calcium, a possibility that could be readily tested using PLC inhibitors. 2) more

problematic, a permissive cytosolic Ca²⁺ concentration is required for the activity of IP₃ receptors due to their bell-shaped calcium (and ATP) dependency. The lack of a Ca²⁺ release component in the absence of external calcium could thus reflect a refractory state of the IPTR that re-opens when calcium is added back as permissive conditions are restored, mimicking the observed tunnelling effect. Have the authors experimental proof that the tunnelling responses do not reflect different sensitivities of the IP₃ receptor instead of differences in the calcium content of intracellular stores? I realise that this might not be an easy experiment but how would using micromolar instead of millimolar calcium concentration impact the tunnelling responses?

Figs 1-5 nicely illustrate and quantify the relative location and redistribution of calcium handling proteins during tunnelling, but do not present fundamentally novel evidence or concepts. I would suggest condensing these data into three main and supplementary figures and expand the subsequent figures reporting the functional effects of the molecular inhibitor. This will fluidify the narrative and strengthen the impact of the study.

Fig. 6: additional controls are required for the validation of the new calcium attenuator. 1) in all the experiment reporting the effect of CaTAr the MAPPER should be used as control since the inhibitors are derived from the MAPPER backbone. 2) The effects of the SERCA inhibitor on the ER refilling kinetics should be documented. Ideally, experiments should be designed to document a local inhibition, for instance by using TIRF imaging of genetically encoded calcium indicators targeted to the ER to document the loss of refilling in cortical ER structures despite preserved global refilling. 3) This figure also contrasts data of very different statistical power. Preserved calcium release and influx is shown in panels 6E and 6G with n=10-20 (cells? recordings? experiments?) while a significant inhibition of calcium tunnelling is shown in Fig. 6K with n=100-270 (likely cells, imaged simultaneously as in 6I). It is inappropriate to compare experiments with a tenfold difference in statistical power. Results from experiments with comparable sample sizes should be presented with the actual P value indicated on every panel and a detailed description of the n number provided for all experiments. In this regard I am not sure that individual cells recorded simultaneously on the same dish can be considered truly independent, especially if the magnitude of the inhibition depends on the levels of the protein expressed, as illustrated in 6J. Reporting the calcium response amplitude as a function of the protein expression levels, derived from its GFP fluorescence would be informative here.

Fig. 6D: The lack of effect of the CaTAr on the Ca²⁺ response evoked by Ca²⁺ readmission to cells treated with CPA is surprising, and at odds with the expected effect of a local inhibition of SERCA at MCS, which should enhance STIM-ORAI interactions. One possibility that has not been explored nor discussed is that the local inhibition of SERCA increases the magnitude of Ca²⁺ microdomains around CRAC channels, thereby promoting their Ca²⁺-dependent inactivation. This could be tested by measuring the rates of Mn²⁺ quenching, applied together with calcium during the protocol of FIG 6D. Such a mechanism would enhance the inhibitory effect of CaTAr on refilling and is conceptually important to validate or exclude. The effect of CaTAr and MAPPER on store-operated calcium entry evoked by the irreversible inhibition of SERCA pumps with thapsigargin should also be tested to exclude SERCA-independent effects.

Fig. 7. It would be nice to establish the relative localization of the effector ANO1 relative to the CaTAr inhibitors that prevent tunnelling. From the data presented it appears that SERCA, ANO1, MAPPER, and CaTAr are located at similar distances from the STIM1 clusters, used as reference markers. If the inhibitors co-localize with the effectors (at the diffraction-limited level of the confocal microscope), then the sketch on Figure 8 should be revised.

Minor issues

Line 166: Fig 2C and 2D both show Z profiles.

Lines 231-38: should mention that the size of the SOCE microdomain is derived from EM data while the distances estimated for STIM1-IPTR are derived from confocal imaging.

Lines 272-75: Should mention that EM evidence was provided for stratification in the cortical ER by Orci et al who reported pre cortical and two types of cortical ER structures. PMID: 9906989

Lines 336-37: "Expression of MAPPER does not significantly affect calcium tunnelling (Fig. 6A-C)" but the slope of the calcium response appears reduced in Fig 6B and is not quantified in 6C. Please present a statistical evaluation also for the slope and revise the statement if necessary.

Line 346-48. What protocol was used to evoke nuclear translocation of NFAT in Fig 6H?

Reviewer #3 (Comments to the Authors (Required)):

The manuscript entitled, "Architecture of Ca²⁺ tunneling, a basic Ca²⁺ signaling modality important for secretion" is an interesting and important study. The authors describe a mechanism whereby Ca²⁺ movements are aided by the ER to reach medium range distances through the coordinated activities of SERCA, IP₃R, STIM/Orai and ANO1. The manuscript is organized in 2 major sections; the first half of the manuscript is primarily focused on describing the 3 dimensional architecture of the distinct

components involved in Ca²⁺ tunnelling. The second half takes advantage of 2 selective inhibitors of Ca²⁺ tunnelling (termed CaTar1 and CaTar2) designed by the authors to establish the implications of Ca²⁺ tunnelling in vitro and in vivo (for sweating). While I found the first half of the manuscript extremely difficult to follow, I was very impressed with the design and utilization of CaTar1 and CaTar2. I really have no major criticisms, but I do think that revision of the first half of the work for readability would be helpful.

Minor Comments:

1. Difficult though it may be, careful rewriting of the results sections describing the first 5 figures for readability will increase the impact of this paper.
2. I had a little more trouble than I should have understanding the assay used to measure calcium tunnelling than I should have. While this strategy has been previously published, it is not mainstream and needs to be justified properly in this study. In 1D and 1E, how do you know that introducing Cch drives calcium current due to tunnelling as opposed to affecting channel activity or releasing more calcium from the ER?

27 June 2024

Re: JCB manuscript #202402107

Response to Reviewers Comments

We thank the Editors for their insights and the Reviewers for their careful and constructive review of our paper, and for their positive comments in support of the manuscript. Below is a point-by-point response to the critiques raised. For each comment the input of the Reviewer is listed in *"italics between quotes"* followed by our response. In the interest of conciseness, only critical comments that require a response are listed.

Editorial Comments

"Reviewers #1&2 are also supportive but request new experiments that we agree would be important to add. These include data showing that activation of ionotropic receptors and different sensitivities of the IP3 receptor do not cause Ca²⁺ release from the ER, additional controls for the CaTAr inhibitors, assays of Ca²⁺ tunnelling under PLC activation, improved statistical analysis, and clarification of several results."

We have performed the requested additional experiments in response to the specific issues raised by each Reviewer as outlined in detail below.

Reviewer #1

"...STIM1 puncta do not overlap with the SERCA2b Ca²⁺ pumps, and the latter is found (enriched?) in the adjacent ER regions."

We have not observed any enrichment per se of the SERCA pump around STIM1 puncta, we rather note that the SERCA pump is excluded from the SOCE microdomain/ERPMCS following store depletion and appears to be distributed throughout the cortical and deep ER.

"These are interesting studies, extending the Authors previous observations. Analysis of the structural arrangement as well as showing the physiological relevance of Ca²⁺ tunneling advance our knowledge on various modes of Ca²⁺ signal propagation. While I appreciate the clever design of the CaTAr construct, I feel that there are important controls missing to validate the conclusions. Moreover, the Authors appear to ignore the important difference between ER store depletion by Tg and by PLC activation, as the latter also generates lipid changes in the PM in addition to generating IP3. I also question some of the conclusions based on the immunostaining of SERCA2b."

Please see our responses to the specific comments raised by the Reviewer below as they address all the issues listed here.

"1. SERCA2b enrichment is not convincing around STIM1 clusters within the ER after Tg treatment. In fact, Fig. S2 shows more SERCA2b clusters in control cells that seem to dissipate

after ER store depletion.”

We agree with the Reviewer’s observation that there is no enrichment of SERCA around STIM1 clusters following store depletion with TG as compared to its distribution in the rest of the ER. Rather we observe that SERCA is excluded from the STIM1 cluster and localizes around it and deeper within the cortical ER. We suspect the confusion arise from our misplaced use of the word enriched on page 7 in the following sentence: “...following store depletion SERCA is not present within STIM1 clusters at ERPMCS, but is rather enriched cortically in their vicinity, both laterally and axially (Fig. 2T).” We have accordingly replaced “is rather enriched cortically” with “rather localizes cortically” in the revised manuscript.

“2. The Authors use thapsigargin (Tg) to activate SOCE in all confocal experiments to show the relative distribution of the putative molecular players. However, “Ca²⁺ tunneling”, by definition, happens when IP₃ is also generated, that is during PLC activation. PLC activation, however, also generates another set of messengers, namely diacylglycerol (DAG) and phosphatidic acid (PA). These messengers would activate other effectors, such as PKC or the P1TP transfer proteins, Nir2/Nir3, which can have an impact on how Ca²⁺ signals are generated or propagated. I feel that this angle of the story is completely overlooked. It would be informative to use caged IP₃ instead of agonist stimulation to assess Ca²⁺ tunneling without the other effects of PLC activation that inevitably adds a further layer of complexity.”

We agree with the Reviewer and thank them for this excellent suggestion. It would indeed be important to rule out a specific role for other lipid second messenger or effectors downstream of PLC activation following Histamine treatment in inducing or modulating Ca²⁺ tunneling. We have used Caged IP₃ as suggested by the Reviewer in additional experiments to address this issue. These experiments are now summarized in a new Figure 5 in the revised manuscript. These experiments show that similar to Histamine addition, uncaging caged IP₃ dramatically stimulates Ca²⁺ tunneling (Fig. 5). That is under the tunneling protocol (store depletion with CPA and wash to restore SERCA function) uncaging IP₃ induces a very large increase in cytosolic Ca²⁺, by activating Ca²⁺ tunneling (Fig. 5). This shows that Ca²⁺ tunneling requires IP₃ receptor gating and does not rely on other lipid messenger that may be generated downstream of PLC activation. As is the case with the tunneling induced by His, tunneling induced by uncaging IP₃ is also inhibited by CatAr1 (Fig. 5D).

3. While the 3D reconstruction of the PM-adjacent ER is impressive, I do not see its value as it relates to the main message of the paper. Perhaps immuno-detection of the SERCA pump in these images relative to STIM1/Orai1 clusters would make this EM analysis a lot more relevant.

We agree with the Reviewer and have accordingly removed these data from the revised manuscript.

4. The design of CaTAr is really clever, but it has some caveats. Namely, this construct has a polybasic domain at the C-terminus that interacts with the negatively charged lipids, PI4P and PI(4,5)P₂ in the plasma membrane. This construct, indeed, is pushed to the periphery of the STIM1/Orai1 clusters upon ER store depletion, when Tg was used to empty the Ca²⁺ stores as its interaction with the PM lipids is still maintained. However, when the cells are activated by a PLC agonist, PI(4,5)P₂ and PI4P levels may drop, which limits the association of the polybasic domain with the PM and could allow CaTAr to diffuse away from STIM1-adjacent ER regions. The Authors

should have shown the localization of this construct (as well as some of the other molecular players) after PLC activation and not only after Tg treatment.

This is again an excellent suggestion to confirm that CaTAr localizes relative to STIM1 in a similar fashion following PLC activation as observed following store depletion with thapsigargin. We have performed these experiments and show that CaTAr localizes to areas surrounding STIM1 clusters following His treatment (Fig. 5E-F), similar to what is observed following store depletion with TG. As expected, given the less pronounced store depletion in response to His as compared to TG the STIM1 clusters formed were smaller. These data are now summarized in the new Figure 5 in the revised manuscript.

5. The Authors used the expression of the full-length PLN protein to show its inhibition of the SERCA pump when expressed. However, a better control would be to show that the CaTAr construct exerts a global SERCA inhibition when released from the PM contacts by removing its polybasic domain.

We agree with the Reviewer that this would be a nice additional control to have and have initiated these experiments but have faced some technical difficulties in generating the CaTAr construct with the polybasic domain deleted. In the interest of a timely revision, we are submitting the revised manuscript without these additional experiments because we have shown that CaTAr specifically inhibits tunneling without affecting SOCE per se or Ca²⁺ release from stores. If the reviewer feels strongly about this control we can add it to the final version of the paper.

Reviewer #2

"The high-quality imaging and innovative functional approaches clearly establish the physiological relevance of calcium tunnelling in fluid secretion. My enthusiasm is somewhat limited by the rather redundant presentation of the first figures and the lack of control experiments for the validation of the genetically encoded SERCA inhibitors. I also have concerns with the statistical analysis as detailed below."

We thank the Reviewer for the positive comments. Regarding the Figures, controls, and statistical analyses please see our detailed responses below.

"The functional evidence that calcium is "tunnelling" relies on the presence of Ca²⁺ responses when agonists that fail to evoke a response in calcium free medium are applied immediately during calcium readmission (Fig. 1 and S1). The lack of response in Ca²⁺-free is taken as proof that the tunnelling response does not reflect calcium release from the ER. I have two concerns here. 1) did the authors exclude the possibility that the agonist could activate ionotropic receptors? These would generate similar responses entirely dependent on external calcium, a possibility that could be readily tested using PLC inhibitors. 2) more problematic, a permissive cytosolic Ca²⁺ concentration is required for the activity of IP3 receptors due to their bell-shaped calcium (and ATP) dependency. The lack of a Ca²⁺ release component in the absence of external calcium could thus reflect a refractory state of the IPTR that re-opens when calcium is added back as permissive conditions are restored, mimicking the observed tunnelling effect. Have the authors experimental proof that the tunnelling responses do not reflect different sensitivities of the IP3 receptor instead of differences in the calcium content of intracellular stores? I realise that this might not be an easy

experiment but how would using micromolar instead of millimolar calcium concentration impact the tunnelling responses?”

- 1) The idea of activation of an ionotropic receptor is interesting and one we hadn't considered before. However, several pieces of evidence argue against that possibility. We are unaware of any Ca^{2+} permeant ionotropic histamine receptor so it is unlikely that histamine would directly activate Ca^{2+} entry. However, it is possible that histamine could be modulating an ionotropic Ca^{2+} entry pathway indirectly. This latter possibility is also unlikely because if Ca^{2+} was entering through an ionotropic receptor it would not be inhibited by CaTAR expression. One would rather predict that CaTAR would enhance an ionotropic entry mechanism. In contrast, as shown in Figure 4, tunneling is inhibited in a dose-dependent fashion by CaTAR expression.
- 2) Regarding the sensitivity of IP_3 receptors to Ca^{2+} , the Reviewer raises an interesting point where Ca^{2+} flowing through SOCE after Ca^{2+} re-addition could in fact be modulating IP_3 receptor gating to release more Ca^{2+} from stores. Although this is an interesting idea, we have previously ruled it out (Courjaret et al. Sci. Reports 8:11214). The tunneling protocol was designed to fully deplete intracellular Ca^{2+} stores as confirmed by a second application of His after the CPA treatment (please see attached Fig. 2D from the Sci Rep paper). Therefore, independent of the sensitivity of the IP_3 receptors to Ca^{2+} and how Ca^{2+} may be affecting their gating they cannot be releasing store Ca^{2+} upon Ca^{2+} re-addition as the store are empty. In addition, for the experiments with the primary salivary gland cells the same control was performed to confirm that the CPA followed by washout treatment fully depleted the stores (Supp. Fig. 1A).

“Figs 1-5 nicely illustrate and quantify the relative location and redistribution of calcium handling proteins during tunnelling, but do not present fundamentally novel evidence or concepts. I would suggest condensing these data into three main and supplementary figures and expand the subsequent figures reporting the functional effects of the molecular inhibitor. This will fluidify the narrative and strengthen the impact of the study. “

We have taken the Reviewer suggestion and consolidated the first 5 figures into 3 figures while removing some of the data into the supplemental figures. This was associated with some changes in the text to better streamline the flow of the paper.

“Fig. 6: additional controls are required for the validation of the new calcium attenuator. 1) in all the experiment reporting the effect of CaTAR the MAPPER should be used as control since the inhibitors are derived from the MAPPER backbone. 2) The effects of the SERCA inhibitor on the ER refilling kinetics should be documented. Ideally, experiments should be designed to document a local inhibition, for instance by using TIRF imaging of genetically encoded calcium indicators targeted to the ER to document the loss of refilling in cortical ER structures despite preserved global refilling. 3) This figure also contrasts data of very different statistical power. Preserved calcium release and influx is shown in panels 6E and 6G with $n=10-20$ (cells? recordings? experiments?) while a significant inhibition of calcium tunnelling is shown in Fig. 6K with $n=100-270$ (likely cells, imaged simultaneously as in 6I). It is inappropriate to compare experiments with a

tenfold difference in statistical power. Results from experiments with comparable sample sizes should be presented with the actual *P* value indicated on every panel and a detailed description of the *n* number provided for all experiments. In this regard I am not sure that individual cells recorded simultaneously on the same dish can be considered truly independent, especially if the magnitude of the inhibition depends on the levels of the protein expressed, as illustrated in 6J. Reporting the calcium response amplitude as a function of the protein expression levels, derived from its GFP fluorescence would be informative here.”

1. We did use MAPPER as the control and showed that it did not inhibit tunneling in Figure 4. We also tried to use it for the CaTAr experiments however there was too much variability in the cytosolic Ca²⁺ signals observed during tunneling between different dishes. This required a tighter internal control for every dish analyzed. We therefore used binned data based on CaTAr expression to illustrate the CaTAr-dependent inhibition of tunneling based on the levels of expression. Binning was based on normalizing the expression levels for each dish to minimize variability in transfection efficiency and expression levels between dishes. We clarify this in the Methods section accordingly in the revised manuscript and now indicate the number of dishes in the legend together with the actual p values. Furthermore, as requested by the Reviewer we illustrate in Supplemental Figure 10A tunneling amplitude as a function of CaTAr expression, which shows a dose-dependent inhibition.

2. We attempted to measure the potential changes in cortical refilling efficiency in cells expressing CaTAr and MAPPER by using the genetically encoded ER Ca²⁺-sensor R-CEPIA ER and imaging the cortical region using TIRF microscopy. As indicated in the attached Figure, we did not detect any change in the refilling kinetics of the ER fraction imaged in the TIRF plane between MAPPER and CaTAr1 expressing cells. This could be because the TIRF layer might still be too thick to discriminate between junctional ER at the SOCE microdomain and deeper ER, which is also cortical in our recording conditions.

3. We agree with the Reviewer that the large difference between the number of experiments in old Figure 6 now Figure 4, limits the statistical power of the data. We therefore performed additional recordings to increase the number of data points. This did not alter our initial observation that there is no significant inhibition of SOCE by CaTAr under the tunneling protocol when a more physiological SOCE is induced with active SERCA pumps.

“Fig. 6D: The lack of effect of the CaTAr on the Ca²⁺ response evoked by Ca²⁺ readmission to cells treated with CPA is surprising, and at odds with the expected effect of a local inhibition of SERCA at MCS, which should enhance STIM-ORAI interactions. One possibility that has not been explored nor discussed is that the local inhibition of SERCA increases the magnitude of Ca²⁺ microdomains around CRAC channels, thereby promoting their Ca²⁺-dependent inactivation. This could be tested by measuring the rates of Mn²⁺ quenching, applied together with calcium during the protocol of FIG 6D. Such a mechanism would enhance the inhibitory effect of CaTAr on refilling and is conceptually important to validate or exclude. The effect of CaTAr and MAPPER on store-

operated calcium entry evoked by the irreversible inhibition of SERCA pumps with thapsigargin should also be tested to exclude SERCA-independent effects.”

We agree with the Reviewer and were surprised as well by the lack of effect of CaTAr on SOCE after CPA washout. One would indeed expect some inhibition of SOCE due to the higher predicted Ca^{2+} in the SOCE microdomain, which would induce inactivation. As requested by the Reviewer we tested the effect of CaTAr on SOCE induced after complete inhibition of SERCA by thapsigargin (new Fig. 4I) and recorded a small reduction in the amplitude of SOCE. However, in this case it is unlikely that the effects of CaTAr on SOCE are due to modulating SERCA activity as it is completely blocked by TG. Rather, this small inhibition may be attributed to effects independent of the inhibition of SERCA but to the perturbation of the cortical domain by the expression of CaTAr as a similar small inhibition of SOCE has been observed following MAPPER expression (Henry et al., 2022).

The lack of effect on SOCE induced after CPA washout following CaTAr expression may be due to the rather small SOCE signal observed under these condition while imaging whole cell Ca^{2+} as most of the signal would be expected to localize to the SOCE microdomain which is poorly resolved in whole cell imaging experiments. Therefore, to get a better more physiological readout of Ca^{2+} levels in the SOCE microdomain with minimal experimental perturbations of intracellular Ca^{2+} levels, we used NFAT activation downstream of calcineurin as an endogenous SOCE microdomain localized effector of Ca^{2+} influx as a readout of integrated SOCE microdomain Ca^{2+} levels (Fig. 4J). As observed with the SOCE measurements following CPA washout we observe no inhibition of NFAT translocation following CaTAr expression. Granted because of the integration of signaling downstream of calcineurin a small inhibition of Ca^{2+} in the SOCE microdomain may be masked in these experiments, yet we feel this is a more relevant measure of microdomain Ca^{2+} than the Mn quench experiment which would measure the rate of flux through the SOCE channel as this would be a very difficult experiment to quantify given the need to include Ca^{2+} be definition to get any Ca^{2+} -dependent inactivation. Although, this does not directly address the Reviewer's concern it argues that if there are any changes to microdomain Ca^{2+} that would affect CRAC inactivation it is rather small.

“Fig. 7. It would be nice to establish the relative localization of the effector ANO1 relative to the CaTAr inhibitors that prevent tunnelling. From the data presented it appears that SERCA, ANO1, MAPPER, and CaTAr are located at similar distances from the STIM1 clusters, used as reference markers. If the inhibitors co localize with the effectors (at the diffraction-limited level of the confocal microscope), then the sketch on Figure 8 should be revised.”

We have performed the colocalization of CaTAr2 and ANO1 as requested by the Reviewer. These new data are illustrated in the new supplemental Figure 11 in the revised manuscript. They show that although there is some diffuse ANO1 that colocalize with CaTAr2, ANO1 is enriched at sites lining the edge of CaTAr2 as well as cell compartments such as membrane protrusions that are very distant from CaTAr2 (please see Supp. Fig. 11).

Minor issues

“Line 166: Fig 2C and 2D both show Z profiles.”

Yes both show Z profiles as one is at the whole cell level and one is at the level of individual

cluster. To avoid confusion we moved the cluster quantification to supplemental figures.

Lines 231-38: should mention that the size of the SOCE microdomain is derived from EM data while the distances estimated for STIM1-IPTR are derived from confocal imaging.

This is a good suggestion. We added a sentence accordingly (line 242-4).

"Lines 272-75: Should mention that EM evidence was provided for stratification in the cortical ER by Orci et al who reported pre cortical and two types of cortical ER structures. PMID: 9906989"

Thanks for another good suggestions. This was added (lines 280-282).

"Lines 336-37: "Expression of MAPPER does not significantly affect calcium tunnelling (Fig. 6A-C)" but the slope of the calcium response appears reduced in Fig 6B and is not quantified in 6C. Please present a statistical evaluation also for the slope and revise the statement if necessary."

As requested by the Reviewer we performed a statistical analysis of the slope of the tunneled Ca²⁺ signal after MAPPER expression and consistent with the individual example shown we do observe a significantly slower rising slope following MAPPER expression (Fig. 4D). This could be a reflection of the expansion of ERPMCS observed following MAPPER expression as discussed in the text of the revised manuscript.

"Line 346-48. What protocol was used to evoke nuclear translocation of NFAT in Fig 6H?"

We activated SOCE using store depletion with thapsigargin to induce NFAT translocation. This is now clarified in the methods.

Reviewer #3

"While I found the first half of the manuscript extremely difficult to follow, I was very impressed with the design and utilization of CaTar1 and CaTar2. I really have no major criticisms, but I do think that revision of the first half of the work for readability would be helpful."

We thank the Reviewer for the positive comments. Regarding the first half of the manuscript (Figures 1-5), we have taken the Reviewer's suggestion by consolidating these figures into 3 figures and moving some of the data to supplemental figures. We also edited the text to simplify the flow of the portion of the paper. Reviewer 2 had a similar concern.

Minor Comments:

"1. Difficult though it may be, careful rewriting of the results sections describing the first 5 figures for readability will increase the impact of this paper."

We agree and have revised both the Figures (Fig. 1-5) and the text to improve readability.

"2. I had a little more trouble than I should have understanding the assay used to measure calcium tunnelling than I should have. While this strategy has been previously published, it is not mainstream and needs to be justified properly in this study. In 1D and 1E, how do you know that introducing Cch drives calcium current due to tunnelling as opposed to affecting channel activity or

releasing more calcium from the ER?"

These are very important points that we have previously ruled out in our previous paper (Courjaret et al. *Sci. Reports* 8:11214). We know that Ca^{2+} is being tunneled because in the tunneling protocol stores are depleted with CPA, which is then washed out to allow for SERCA to be active. We confirmed that the store were depleted using a second application of His after the CPA treatment (please see attached Fig. 2D from the Sci Rep paper). So after extracellular Ca^{2+} addition, pre-existing Ca^{2+} in the stores cannot be the source of the observed Ca^{2+} signal. In addition, for the experiments with the primary salivary gland cells the same control was performed to confirm that the CPA followed by washout treatment fully depleted the stores (Supp. Fig. 1A).

As discussed above in response to a question raised by Reviewer 1, another possibility is that His is activating ionotropic receptors or modulating Ca^{2+} influx channels when added together with Ca^{2+} . This is unlikely however, because if this was the case it would not be blocked by the CaTAR inhibitors rather one would expect a larger signal.

References

Henry, C., A. Carreras-Sureda, and N. Demareux. 2022. Enforced tethering elongates the cortical endoplasmic reticulum and limits store-operated Ca^{2+} entry. *J. Cell Sci.* 135.

August 22, 2024

Re: JCB manuscript #202402107R

Prof. Khaled Machaca
Weill Cornell Medical College in Qatar
Luqta Street
Education City
Doha 24144
Qatar

Dear Prof. Machaca,

Thank you for submitting your revised manuscript entitled "Architecture of Ca²⁺ tunneling, a basic Ca²⁺ signaling modality important for secretion." The manuscript has been seen by two of the original reviewers whose full comments are appended below. While the reviewers continue to be overall positive about the work in terms of its suitability for JCB, some important issues remain.

You will see that the reviewers feel your study has been significantly improved. However, both ask for verification that CaTAr is a SERCA inhibitor, if that is feasible. Rev #2 has some additional concerns that you can address experimentally or in the text.

Our general policy is that papers are considered through only one revision cycle; however, given that the suggested changes are relatively minor we are open to one additional short round of revision. Please note that I will expect to make a final decision without additional reviewer input upon resubmission.

Please submit the final revision within 3 months, along with a cover letter that includes a point by point response to the remaining reviewer comments.

Thank you for this interesting contribution to Journal of Cell Biology. You can contact me or the scientific editor listed below at the journal office with any questions at cellbio@rockefeller.edu.

Sincerely,

William Prinz, PhD
Monitoring Editor
Journal of Cell Biology

Dan Simon, PhD
Scientific Editor
Journal of Cell Biology

Reviewer #1 (Comments to the Authors (Required)):

This is a revised version of a previously reviewed manuscript.

The Authors have addressed most of my comments and concerns. I still would have liked to see the control experiment testing the inhibitory effect of the CaTAr construct on SERCA activity (not only that of the wild-type PLN). While I would not hold the paper back on this single (although critically important) point, if there is a second revision, I would certainly like the Authors to address this comment.

Overall, the revised version has greatly improved and made the paper more competitive for JCB.

Reviewer #2 (Comments to the Authors (Required)):

The authors have performed additional experiments to address the points raised by the reviewers. The experiments with InsP3 uncaging are particularly interesting as they nicely corroborate the findings obtained with histamine, ruling out an implication of other mediators generated by PLC activation in the tunneling process. I find however that my specific queries regarding the tunneling protocol and the SERCA inhibition by CaTAr. below were not properly addressed and would ask for additional evidence or reconsideration.

1. Implication of ionotropic receptors. The authors argue that this is unlikely because such an effect should not be inhibited but enhanced by their CaTar inhibitor. This is a bit of a circular argument that assumes that the inhibitor is acting exclusively against its intended target. As it stands the inhibition of local SERCA pumping by the new inhibitor has not been directly demonstrated (see point 3 below). A more direct proof using inhibitors of purinergic ionotropic receptors would be more appropriate here. Alternatively, they could show pharmacologically or genetically that only Orai1 channels fuel the tunneling response or refer to prior publication documenting this point.

2. Refractory state of InsP3 receptors. Here they argue that this mechanism is unlikely because they showed in an earlier publication that addition of histamine in Ca²⁺-free medium fails to further deplete the ER after transient CPA exposure. This evidence is however inconclusive because the lack of response could be precisely due to a refractory state of InsP3 receptors in these conditions, which is the possibility that I would like to exclude. More direct evidence that the stores are entirely depleted as claimed could be obtained using ionomycin instead of histamine in this experiment. The authors may have already generated these data that would further validate the tunneling protocol.

3. Validation of CaTar1 as bona fide SERCA inhibitor. As requested, the authors have compared the efficiency of ER Ca²⁺ refilling in cells expressing MAPPER or CaTar1, using R-CEPIA-ER and TIRF microscopy. These new data show that the ER refilling kinetics measured in the TIRF plane are not impacted by CaTar1. I am surprised by these data which argue against a local inhibition of SERCA at ER-PM contact sites. As normalized data are shown, it is unclear whether the magnitude of ER refilling was comparable between the two conditions. Could the author provide the original, non-normalized data? Or repeat this experiment with a lower Ca²⁺ concentration, to maximize the differences in refilling? Or remobilize the Ca²⁺ repumped in the presence and absence of CaTar1 by readding a SERCA inhibitor in Ca²⁺-free after a short refilling period? In the absence of proof that SERCA are locally inhibited by CaTar1, one cannot exclude that CaTar is impacting tunneling indirectly, for instance by inhibiting or displacing IP3 receptors. These possibilities should be excluded or mentioned.

Line 203 typo "form" = from

29 September 2024

Re: JCB manuscript #202402107

Response to Reviewers Comments

We thank the Editors and the Reviewers for their constructive review and support of our paper. We have addressed the one remaining issue requested by the Reviewers that is a control experiment showing that the CaTAr inhibitor blocks SERCA function. These new data are now added to the revised manuscript. Below is a point-by-point response to the critiques raised. For each comment the input of the Reviewer is listed in *“italics between quotes”* followed by our response. In the interest of conciseness, only critical comments that require a response are listed.

Editorial Comments

“You will see that the reviewers feel your study has been significantly improved. However, both ask for verification that CaTAr is a SERCA inhibitor, if that is feasible. Rev #2 has some additional concerns that you can address experimentally or in the text.”

We have generated a CaTAr version lacking the poly-lysine domain that targets it to ERPMCS. This construct CaTAr Δ polyK localizes broadly to the ER. We show that CaTAr Δ polyK inhibits SERCA globally resulting in store depletion at rest. These new experiments are added to the manuscript as Supplemental Figure 8E-H. We further address the additional concerns raised by Reviewer 2 as outline below.

Reviewer #1

“The Authors have addressed most of my comments and concerns. I still would have liked to see the control experiment testing the inhibitory effect of the CaTAr construct on SERCA activity (not only that of the wild-type PLN). While I would not hold the paper back on this single (although critically important) point, if there is a second revision, I would certainly like the Authors to address this comment.”

We agree with the Reviewer that this is a critical control. To directly show that CaTAr inhibits SERCA activity, we generated a CaTAr1 construct lacking the N-terminal poly-lysine domain that targets it to ERPMCS, referred to as CaTAr Δ polyK. As expected, CaTAr Δ polyK localizes broadly to the ER as shown in the new Supplemental Figure 8E, and its expression leads to store depletion resulting in the absence of Ca²⁺ release in response to histamine (Supp. Fig. 8F-H). These results replicates what is observed with the expression of PLN alone (Supplemental Fig. 8A-D). This shows that CaTAr through its embedded PLN domain directly inhibits SERCA activity as revealed by the global store depletion in response to CaTAr Δ polyK expression due to the passive ER Ca²⁺ leak in the absence of SERCA activity.

Reviewer #2

1. *“Implication of ionotropic receptors. The authors argue that this is unlikely because such an effect should not be inhibited but enhanced by their CaTar inhibitor. This is a bit of a circular argument that assumes that the inhibitor is acting exclusively against its intended target. As it*

stands the inhibition of local SERCA pumping by the new inhibitor has not been directly demonstrated (see point 3 below). A more direct proof using inhibitors of purinergic ionotropic receptors would be more appropriate here. Alternatively, they could show pharmacologically or genetically that only Orai1 channels fuel the tunneling response or refer to prior publication documenting this point.”

As discussed in our previous response; the potential involvement of ionotropic receptors in tunneling is unlikely and does not fit the current experimental data. As previously mentioned, we are unaware of any histamine ionotropic receptors, and experimentally we are activating tunneling using histamine to produce IP3 and gate IP3Rs, so it is not clear how the Reviewer proposes that histamine activates ionotropic receptors to directly mediate Ca²⁺ entry? Second, if ionotropic receptors were involved, tunneling would not depend on SERCA or on Ca²⁺ store content as we have shown in our previous papers (Courjaret et al., 2018; Courjaret and Machaca, 2014). We further show in the current manuscript that tunneling requires SERCA activity as it is inhibited by CaTAr and we now show that CaTAr directly inhibits SERCA (Supp. Fig. 8E-H).

Please see the below Figures from Courjaret et al. 2018. *Panel A*: the 2nd His addition -when the store are full- leads to Ca²⁺ release and Ca²⁺ store depletion as seen in the ER Ca²⁺ content trace (red). *Panel B*: the 1st His addition when ER Ca²⁺ stores are depleted (red trace no Ca²⁺ in the ER) does not lead to a cytosolic Ca²⁺ response. If His activates ionotropic receptor a cytosolic Ca²⁺ rise is expected even when ER Ca²⁺ stores are depleted. Collectively, these data show that His does not activate ionotropic receptors and that tunneling requires SERCA activity.

2. “Refractory state of InsP3 receptors. Here they argue that this mechanism is unlikely because they showed in an earlier publication that addition of histamine in Ca²⁺-free medium fails to further deplete the ER after transient CPA exposure. This evidence is however inconclusive because the lack of response could be precisely due to a refractory state of InsP3 receptors in these conditions, which is the possibility that I would like to exclude. More direct evidence that the stores are entirely depleted as claimed could be obtained using ionomycin instead of histamine in this experiment. The authors may have already generated these data that would further validate the tunneling protocol.”

The suggested ionomycin experiment would not be a conclusive approach, as ionomycin could release Ca²⁺ from intracellular Ca²⁺ stores other than the ER. Please refer to the Figures above

from Courjaret et al. 2018, which show conclusively that IP3 Receptors are not in a refractory phase. *Panel A*: the 1st His addition leads to Ca²⁺ tunneling with the ER Ca²⁺ stores fully depleted (see red trace reporting ER Ca²⁺ content using CEPIA ER). The 2nd His addition a couple of minutes later, when Ca²⁺ stores have now refilled, results in robust Ca²⁺ release showing that the IP3Rs are active and not refractory. *Panel B*: after store depletion with CPA, the 1st His addition does not lead to a Ca²⁺ transient. The stores are indeed empty as reported by the ER Ca²⁺ trace (red). This a direct report of ER Ca²⁺ content under this experimental paradigm proving that the store are depleted of Ca²⁺. Collectively, these data show that the IP3Rs are not in a refractory state.

3. *“Validation of CaTAR1 as bona fide SERCA inhibitor. As requested, the authors have compared the efficiency of ER Ca²⁺ refilling in cells expressing MAPPER or CaTAR1, using R-CEPIA-ER and TIRF microscopy. These new data show that the ER refilling kinetics measured in the TIRF plane are not impacted by CaTAR1. I am surprised by these data which argue against a local inhibition of SERCA at ER-PM contact sites. As normalized data are shown, it is unclear whether the magnitude of ER refilling was comparable between the two conditions. Could the author provide the original, non-normalized data? Or repeat this experiment with a lower Ca²⁺ concentration, to maximize the differences in refilling? Or remobilize the Ca²⁺ repumped in the presence and absence of CaTAR1 by readding a SERCA inhibitor in Ca²⁺-free after a short refilling period? In the absence of proof that SERCA are locally inhibited by CaTAR1, one cannot exclude that CaTAR is impacting tunneling indirectly, for instance by inhibiting or displacing IP3 receptors. These possibilities should be excluded or mentioned.”*

We agree with the Reviewer that showing that CaTAR inhibits SERCA is a critical control. To directly show that CaTAR inhibits SERCA activity, we generated a CaTAR construct lacking the N-terminal poly-lysine domain that targets it to ERPMCS, referred to as CaTAR Δ polyK. As expected, CaTAR Δ polyK localizes broadly to the ER as shown in the new Supplemental Figure 8E-H, and its expression leads to store depletion resulting in the absence of Ca²⁺ release in response to histamine. This result replicates what is observed with the expression of PLN alone (Supplemental Fig. 8). This shows that CaTAR through its embedded PLN domain directly inhibits SERCA activity as revealed by the global store depletion in response to CaTAR Δ polyK expression due to the passive ER Ca²⁺ leak in the absence of SERCA activity.

Line 203 typo "form" = from"

Corrected thank you.

Courjaret, R., M. Dib, and K. Machaca. 2018. Spatially restricted subcellular Ca(2+) signaling downstream of store-operated calcium entry encoded by a cortical tunneling mechanism. *Sci. Rep.* 8:11214.

Courjaret, R., and K. Machaca. 2014. Mid-range Ca²⁺ signalling mediated by functional coupling between store-operated Ca²⁺ entry and IP₃-dependent Ca²⁺ release. *Nat Commun.* 5:3916.

October 1, 2024

RE: JCB Manuscript #202402107RR

Prof. Khaled Machaca
Weill Cornell Medical College in Qatar
Research
Luqta Street
Education City
Doha 24144
Qatar

Dear Prof. Machaca,

Thank you for submitting your revised manuscript entitled "Architecture of Ca²⁺ tunneling, a basic Ca²⁺ signaling modality important for secretion." We would be happy to publish your paper in JCB pending final revisions necessary to meet our formatting guidelines (see details below).

A. MANUSCRIPT ORGANIZATION AND FORMATTING:

1) Text limits: Character count for Articles is < 40,000, not including spaces. Count includes title page, abstract, introduction, results, discussion, and acknowledgments. Count does not include materials and methods, figure legends, references, tables, or supplemental legends.

2) Figure formatting: Articles may have up to 10 main text figures. Scale bars must be present on all microscopy images, including inset magnifications. Please add scale bars to Figures S4A, S5E, S7D, & S8B.

Also, please avoid pairing red and green for images and graphs to ensure legibility for color-blind readers. If red and green are paired for images, please ensure that the particular red and green hues used in micrographs are distinctive with any of the colorblind types. If not, please modify colors accordingly or provide separate images of the individual channels.

3) Statistical analysis: Error bars on graphic representations of numerical data must be clearly described in the figure legend. The number of independent data points (n) represented in a graph must be indicated in the legend. Please, indicate whether 'n' refers to technical or biological replicates (i.e. number of analyzed cells, samples or animals, number of independent experiments). If independent experiments with multiple biological replicates have been performed, we recommend using distribution-reproducibility SuperPlots (please see Lord et al., JCB 2020) to better display the distribution of the entire dataset, and report statistics (such as means, error bars, and P values) that address the reproducibility of the findings.

Statistical methods should be explained in full in the materials and methods. For figures presenting pooled data the statistical measure should be defined in the figure legends. Please also be sure to indicate the statistical tests used in each of your experiments (both in the figure legend itself and in a separate methods section) as well as the parameters of the test (for example, if you ran a t-test, please indicate if it was one- or two-sided, etc.). Also, if you used parametric tests, please indicate if the data distribution was tested for normality (and if so, how). If not, you must state something to the effect that "Data distribution was assumed to be normal but this was not formally tested."

4) The title should be concise but accessible to a general readership. To increase the accessibility of the work for a broad audience and non-experts we suggest revising the title to: "The architecture of Ca²⁺ tunneling reveals it is a basic signaling modality important for secretion."

5) Materials and methods: Should be comprehensive and not simply reference a previous publication for details on how an experiment was performed. Please provide full descriptions (at least in brief) in the text for readers who may not have access to referenced manuscripts. The text should not refer to methods "...as previously described."

6) For all cell lines, vectors, constructs/cDNAs, etc. - all genetic material: please include database / vendor ID (e.g., Addgene, ATCC, etc.) or if unavailable, please briefly describe their basic genetic features, even if described in other published work or gifted to you by other investigators (and provide references where appropriate). Please be sure to provide the sequences for all

of your oligos: primers, si/shRNA, RNAi, gRNAs, etc. in the materials and methods. You must also indicate in the methods the source, species, and catalog numbers/vendor identifiers (where appropriate) for all of your antibodies, including secondary. If antibodies are not commercial, please add a reference citation if possible.

7) Microscope image acquisition: The following information must be provided about the acquisition and processing of images:

- a. Make and model of microscope
- b. Type, magnification, and numerical aperture of the objective lenses
- c. Temperature
- d. Imaging medium
- e. Fluorochromes
- f. Camera make and model
- g. Acquisition software
- h. Any software used for image processing subsequent to data acquisition. Please include details and types of operations involved (e.g., type of deconvolution, 3D reconstitutions, surface or volume rendering, gamma adjustments, etc.).

8) References: There is no limit to the number of references cited in a manuscript. References should be cited parenthetically in the text by author and year of publication. Abbreviate the names of journals according to PubMed.

9) Supplemental materials: Articles generally may have up to 5 supplemental figures and 10 videos. You currently exceed this limit and while, in this case, we will be able to give you the extra space, please try to consolidate the supplemental figures if possible. Please also note that tables, like figures, should be provided as individual, editable files. A summary of all supplemental material should appear at the end of the Materials and methods section. Please include one brief sentence per item.

10) Video legends: Should describe what is being shown, the cell type or tissue being viewed (including relevant cell treatments, concentration and duration, or transfection), the imaging method (e.g., time-lapse epifluorescence microscopy), what each color represents, how often frames were collected, the frames/second display rate, and the number of any figure that has related video stills or images.

11) eTOC summary: A ~40-50 word summary that describes the context and significance of the findings for a general readership should be included on the title page. The statement should be written in the present tense and refer to the work in the third person. It should begin with "First author name(s) et al..." to match our preferred style.

13) A separate author contribution section is required following the Acknowledgments in all research manuscripts. All authors should be mentioned and designated by their first and middle initials and full surnames. We encourage use of the CRediT nomenclature (<https://casrai.org/credit/>).

14) ORCID IDs: ORCID IDs are unique identifiers allowing researchers to create a record of their various scholarly contributions in a single place. Please note that ORCID IDs are required for all authors. At resubmission of your final files, please be sure to provide your ORCID ID and those of all co-authors.

15) Journal of Cell Biology now requires a data availability statement for all research article submissions. These statements will be published in the article directly above the Acknowledgments. The statement should address all data underlying the research presented in the manuscript. Please visit the JCB instructions for authors for guidelines and examples of statements at (<https://rupress.org/jcb/pages/editorial-policies#data-availability-statement>).

B. FINAL FILES:

****It is JCB policy that if requested, original data images must be made available to the editors. Failure to provide original images upon request will result in unavoidable delays in publication. Please ensure that you have access to all original data images prior to final submission.****

****The license to publish form must be signed before your manuscript can be sent to production. A link to the electronic license to publish form will be sent to the corresponding author only. Please take a moment to check your funder requirements before choosing the appropriate license.****

Thank you for your attention to these final processing requirements. Please revise and format the manuscript and upload materials within 7 days. If you need an extension for whatever reason, please let us know and we can work with you to determine a suitable revision period.

Thank you for this interesting contribution, we look forward to publishing your paper in Journal of Cell Biology.

Sincerely,

William Prinz, PhD
Monitoring Editor
Journal of Cell Biology

Dan Simon, PhD
Scientific Editor
Journal of Cell Biology